# Ice crystal complexity leads to weaker ice cloud radiative heating in idealized single-column simulations

Edgardo I. Sepulveda Araya[1,*], Sylvia C. Sullivan[1,2,*], and Aiko Voigt[3]

[1]Department of Chemical and Environmental Engineering, University of Arizona, Tucson, AZ 85721, USA.
[2]Department of Hydrology and Atmospheric Sciences, University of Arizona, Tucson, AZ 85721, USA.
[3]Department of Meteorology and Geophysics, University of Vienna, Vienna, Austria.

**Correspondence:** Edgardo I. Sepulveda Araya (edgardo@arizona.edu) and Sylvia C. Sullivan (sylvia@arizona.edu)

**Abstract.** Ice clouds play an important role in the atmospheric radiation budget, both by reflecting shortwave radiation and absorbing or emitting longwave radiation. These effects can modulate the cloud radiative heating (CRH) rate, which in turn influences circulation and precipitation. Ice cloud radiative properties depend on the size, shape (or habit), and complexity, including surface roughness or hollowness, of in-cloud ice crystals. To better predict ice-cloud radiative effects, there has been a continuous effort to account for more ice crystal habits and complexity in current radiative transfer calculations. Here, we conduct a series of idealized single-column radiative transfer calculations to study how ice-CRH responds to including ice crystal complexity. We evaluate four ice optical schemes for a range of ice cloud formation temperatures or altitudes, geometrical depths, ice water paths (IWP), and ice crystal effective radii. In addition, we present a heating rate sensitivity matrix as a condensed visualization of the CRH response across a broad parameter space. We find that including ice complexity in cold thin clouds with high IWP, can diminish the net in-cloud heating and cloud-top cooling by 2.5 K d$^{-1}$ and 15 K d$^{-1}$, respectively. Furthermore, while temperature-based schemes behave similarly to other schemes at warmer temperatures, they predict net CRH at cloud bottom more than 10 K d$^{-1}$ higher than size-dependent schemes at the coldest temperatures. Either weakening of CRH by ice complexity or strengthening by temperature-dependent schemes can alter anvil cloud lifetime and evolution, as well as large-scale atmospheric circulation.

## 1 Introduction

Clouds have a strong influence on Earth's radiative balance, due to their ability to either scatter or absorb the shortwave radiation (SW) coming from the sun and the longwave radiation (LW) coming from the Earth's surface. The resulting radiation balance, not only at the top of the atmosphere (TOA) or the Earth's surface, but also within the atmosphere, determines temperature, pressure gradients and static stability in the atmosphere and thus affects circulation and precipitation patterns (Voigt et al., 2021; Medeiros et al., 2021; Lin and Mapes, 2004).

Both SW and LW cloud radiative effects (CRE), defined as the difference between all-sky and clear-sky net radiative fluxes (downward minus upward) are the most common way to quantify the radiative impact of clouds. While many studies have evaluated TOA or surface CRE, less attention has been paid to the vertical distribution of radiative flux divergences in the atmosphere (Luo et al., 2023; Mace and Wrenn, 2013; Yang et al., 2012; Yan et al., 2016). This greater attention to TOA or

surface CRE is due in part to more abundant satellite and ground-based measurements for validation; relatively few observational datasets exist for in-atmosphere radiative fluxes (Dolinar et al., 2019; Cesana et al., 2019; Kang et al., 2020). Here, we define the cloud-radiative heating rate (CRH) as follows:

$$CRH \equiv -\frac{g_0}{C_P}\frac{d(CRE)}{dP} \tag{1}$$

where $g_0$ is gravitational acceleration and $C_P$ is the specific heat capacity of dry air at constant pressure. In this case, the CRE is evaluated for each atmospheric layer with pressure difference $dP$.

There are a number of motivations to study CRH besides CRE. CRH is not close to zero for ice clouds, as TOA CRE is (Hartmann, 2016). CRH is more directly linked to large-scale circulation, stability, and precipitation, for example through the temperature gradient caused by in-cloud heating and cloud-top cooling in ice clouds (Haslehner et al., 2024; Mather et al., 2007; Tao et al., 1996). A growing body of work highlights the variety of ways in which in-cloud-radiative interactions influence circulation features from the jet stream and extratropical baroclinicity to the North Atlantic Oscillation and El Niño Southern Oscillation (Cesana et al., 2019; Albern et al., 2019, 2021; Li et al., 2015; Papavasileiou et al., 2020; Rädel et al., 2016). At more local scales, these interactions also determine tropical and extratropical cyclone development (Schäfer and Voigt, 2018; Muller and Romps, 2018; Yang and Tan, 2020; Keshtgar et al., 2023; Voigt et al., 2023). Uncertainties in cloud-radiation interactions also propagate to uncertainties in the equilibrium climate sensitivity (on Climate Change , IPCC; Jahangir et al., 2021; Satoh et al., 2018). Finally, CRH profiles will change under global warming, through shifts in the latitudinal gradient in surface shortwave heating, in the tropopause layer longwave heating, and according to the proportionally higher anvil temperature hypothesis (Voigt et al., 2021; Sokol et al., 2024; Zelinka and Hartmann, 2010).

Ice cloud radiative impacts are particularly difficult to constrain because of ice crystal non-sphericity, non-constant densities, and uncertainties in ice formation processes. In-situ data have also revealed that various forms of *ice crystal complexity*— including surface roughness, aerosol or gas inclusions, and microfacets—are pervasive in atmospheric ice crystals (Schnaiter et al., 2016; Magee et al., 2014; Järvinen et al., 2023). Including a more realistic description of ice crystal complexity and habits in atmospheric simulations affects radiative fluxes, by, for example, inducing SW cooling at the TOA (Järvinen et al., 2018) and decreasing land surface temperature (Yi, 2022).

The majority of global climate models (GCMs) and numerical weather prediction (NWP) models make simplifying assumptions, including spherical ice crystals, in their ice optical schemes. However, other more sophisticated optical schemes have been developed, either on the basis of field campaign data or optical particle models. Each of these schemes specifies spectrally resolved cloud optical properties, including the extinction coefficient $\beta_e$, the single scattering albedo $\omega_0$ (SSA), and the asymmetry parameter $g$ as a function of ice crystal effective radius. Different ice optical schemes may employ field campaign data for which measured sizes and habits are quite different, or they may employ optical particle models with a wide range of size and habit assumptions. Ice optical properties and radiative outputs can therefore vary widely from one scheme to the next. The Fu scheme was one of the earliest of these schemes (Fu, 1996; Fu et al., 1998) and is now widely used (Hogan and Bozzo, 2018; Emde et al., 2016; Morcrette et al., 2008), although it assumes that all ice crystals are hexagonal columns. New attempts

to include ice crystal complexity are being proposed, for example taking into account ice crystal surface roughness, which can increase the global SW CRE by 2 W m$^{-2}$ at TOA (Yi et al., 2013).

Previous work has shown how different ice cloud microphysical schemes and processes in storm-resolving models can strongly alter the CRH profile (Sullivan and Voigt, 2021; Sullivan et al., 2022, 2023). The influence of ice cloud processes –and associated uncertainty– on CRH can also vary across global scale climate models (Voigt et al., 2024). Some studies have also looked at the effect of ice optical schemes. For example, Zhao et al. (2018) showed how the use of different optical schemes resulted in CRH differences of 0.2 K day$^{-1}$ near the tropical tropopause. Keshtgar et al. (2024) also identify ice optical schemes as a primary source of uncertainty in potential vorticity generation as extratropical cyclones develop.

We build upon these results here to understand how ice crystal habits and complexity in different optical schemes can impact simulated CRH. In particular, we run four tests using offline single-column radiative transfer calculations and a variety of idealized input ice cloud profiles. These four tests focus on 1) cloud temperature, 2) cloud geometrical depth, 3) ice water path (IWP), and 4) ice crystal effective radius. They also employ four optical schemes of varying sophistication: the Fu scheme that assumes hexagonal crystals (Fu96 for SW and Fu98 for LW (Fu, 1996; Fu et al., 1998)), the Yi13 scheme that assumes an array of habits and prescribes surface roughness (Yi et al., 2013), Baran14 that assumes an array of aggregates (Baran et al., 2014b), and Baran16 that assumes an array of aggregates as well as temperature dependence (Baran et al., 2016). Both Baran's schemes prescribe surface roughness influence in the SW component. These schemes are described in more detail below in Sect. 2.1. We begin by outlining our model setup and inputs in Sect. 2.2. We then present an inter-scheme comparison for each experiment in Sect. 3, and discussion and analysis of these results in Sect. 4.

## 2 Methods

### 2.1 ecRad Radiative Transfer Scheme and Optical Schemes

We use the ecRad radiative transfer model (RTM) of Hogan and Bozzo (2018) version 1.5.0, which is the current operational radiation scheme in the European Centre for Medium-Range Weather Forecasts (ECMWF) Integrated Forecasting System and in the German Weather Service (Deutsche Wetterdienst, DWD) ICON model. ecRad is a two-stream RTM using the same 14 SW and 16 LW spectral bands as the Rapid Radiative Transfer Model (RRTM). We use the Triplecloud solver to avoid stochastic noise in heating rates, a common problem in Monte Carlo radiative transfer solvers. Output files from ecRad contain upward and downward fluxes at each vertical level, both for SW and LW components, and clear- and all-sky conditions, allowing the calculation of heating rates according to Eq. (1). The range of settings and parameters within ecRad makes it a highly modular RTM.

ecRad includes four ice optical scheme options that take the ice mass mixing ratio, effective radius, and temperature as inputs. These optical schemes parameterize the mass extinction coefficient $K_{ext}$, single scattering albedo $\omega_0$, and asymmetry parameter $g$ as a function of the inputs mentioned above. The mass extinction coefficient $K_{ext}$ is the result of both, mass absorption coefficient $K_{abs}$ and mass scattering coefficient $K_{sca}$. $K_{abs}$ is computed as the product between $K_{ext}$ and the co-albedo, or amount of absorbed to total attenuated radiation ($1 - \omega_0$), then quantifying how much radiation is absorbed and how

**Table 1.** Main characteristics of each ice optical scheme. Habits include solid hexagonal columns (Col), bullet rosettes (Bull), solid column aggregates (ColAgg), hollow columns (hCol), hollow bullet rosettes (hBull), droxtals (Drox), plates (Plat), small and large plate aggregates (s,l-PlatAgg). $r_{eff}$ is the effective radius parameter, $q_i$ is the ice mass mixing ratio, and $T$ is temperature. Baran14 has been kept as an experimental, not officially implemented scheme in ecRad.

| Ice Optical Scheme | Habits | Surface Roughness | Inputs |
|---|---|---|---|
| Fu | Col | No roughness | $r_{eff}$ |
| Baran14 | Col, Bull, ColAgg | SW only | $q_i$ |
| Yi13 | Col, Bull, ColAgg, hCol, hBull, Drox, Plat, sPlatAgg, lPlatAgg | Both SW and LW | $r_{eff}$ |
| Baran16 | Col, Bull, ColAgg | SW only | $q_i, T$ |

much is scattered through an ice cloud. Finally, $g$ quantifies the amount of forward versus backward scattered radiation by ice crystals.

The default scheme is the Fu scheme, which includes two parameterizations for the SW and LW components, as functions of the ice crystal effective radius (Fu, 1996; Fu et al., 1998). The Fu scheme was developed using ray-tracing calculations in the SW and Mie scattering theory in the LW and assumes an ensemble of hexagonal ice crystals with sizes based on airborne measurements. Next, the Yi13 scheme evaluates optical properties as a function of ice crystal effective radius, based on a general mixture scheme of 9 ice crystal habits (4 different geometries, plus two including hollowness, and three types of aggregates), together with surface roughness (Yi et al., 2013; Yang et al., 2013; Baum et al., 2011). The interscheme difference in the optical properties between Yi13 and Fu is shown in Fig. 1. $K_{abs}^{SW}$ is 18% smaller in Yi13 than Fu for the effective radius tested in most of our simulations (30 $\mu$m, as described in Sect. 2.2), as a result of smaller $K_{ext}^{SW}$ and larger $\omega_0^{SW}$. In other words, the differences in optical properties result in weaker absorption in the Yi13 scheme. The dependence of optical properties on the effective radius for each scheme is shown in Figs. S1 and S2 of the supplement, for both SW and LW, respectively.

Finally, the Baran schemes use an ensemble of ice columns, bullet rosettes, and column aggregates (Baran et al., 2014b, 2016; Baran and Labonnote, 2007; Baran et al., 2014a), as a function of ice mass mixing ratio. Surface roughness is included in the SW parameterization. The bulk optical properties are computed for a habit ensemble instead of a particle size distribution. The size of the crystals is represented indirectly through ice complexity, where single-column and bullet-rosettes are the smaller crystals, and higher aggregate levels of hexagonal columns are the bigger crystals in the ensemble. In order to correct a temperature and relative humidity bias at the tropical tropopause layer (TTL) from the 2014 scheme when it was used in the UK Met Office model, Baran et al. (2016) updated the scheme to include temperature dependence in optical properties. While Baran14 is implemented as an experimental parameterization in ecRad, we include tests with this scheme to study the effect of this temperature dependence. These four schemes constitute a hierarchy with Fu as the simplest scheme, followed by Baran14, Yi13, and finally Baran16 (see Table 1).

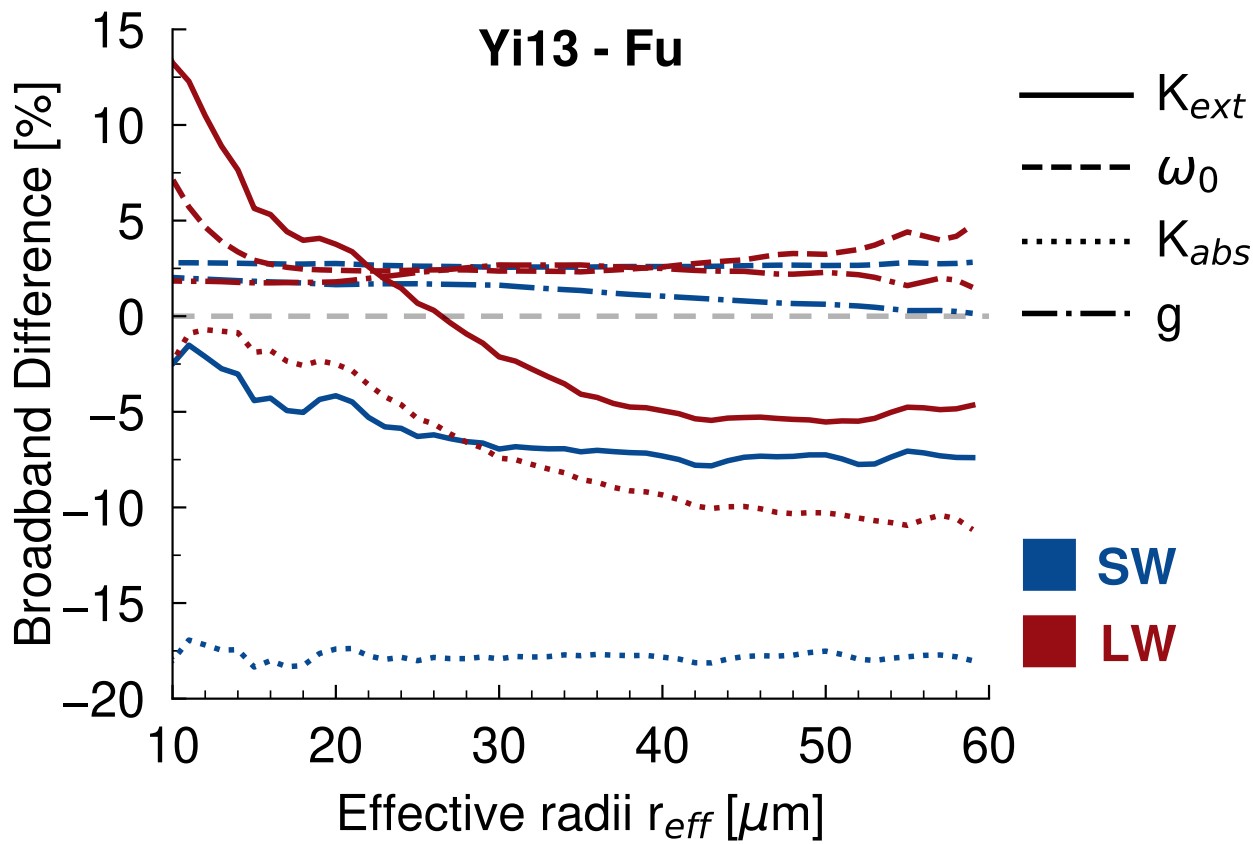

**Figure 1.** Broadband relative difference of mass extinction coefficient $K_{ext}$ (solid line), single scattering albedo $\omega_0$ (dashed line), mass absorption coefficient $K_{abs}$ (dotted line) and asymmetry parameter $g$ (dotted-dashed line), between Yi13 and Fu optical parameterization, as a function of ice crystal effective radii $r_{eff}$. Both radiative components, SW and LW, are shown in blue and red, respectively. Spectrally resolved and absolute values for each scheme are shown in Figs. S1 and S2 of the supplement.

## 2.2 Input Parameters

Climatological profiles of tropical pressure, temperature, and specific humidity are used as inputs to ecRad, along with a skin temperature of 300 K, applicable to tropical oceans (Luo and Minnett, 2020), a SW surface albedo of 0.05 (Hartmann and
115 Berry, 2017), a LW surface emissivity of 0.97, and an effective solar zenith angle (SZA) of $\theta = 53°$ (cosine of SZA $\mu$ of 0.6). A cloud fraction of one is used when the ice cloud layer is present. Climatological values for $O_2$, $CO_2$, $CH_4$ and $N_2O$ mass mixing ratio are used. No cloud liquid water or aerosols are included.

Five series of single-column ice cloud atmospheric profiles are input to ecRad (Table 2). The first set of simulations (Test 1) tests how CRH changes for cloud layers at varying altitudes and corresponding tropical temperatures from 10 km (236
120 K) up to 15.5 km (201 K). For this test, we fix the cloud depth at 1.5 km and effective radius at 30 $\mu$m, following previous

**Table 2.** Variable input parameters and settings used in the idealized single-column simulations and ecRad. The temperature values shown in the second column correspond to mid-cloud temperature, except in Test 2a and Test 2b, where the numbers indicate cloud bottom and cloud top temperature ranges, respectively.

| | *Temperature [K]* | *Geometrical Depth [km]* | *IWP [g m$^{-2}$]* | *Effective radii [$\mu$m]* |
|---|---|---|---|---|
| Test 1 | **236 - 201** | 1.5 | 30 | 30 |
| Test 2a | 236 - 206 ($T_{bottom}$) | **0.5 - 5** | 30 | 30 |
| Test 2b | 233 - 203 ($T_{top}$) | **0.5 - 5** | 30 | 30 |
| Test 3 | 236, 218, 201 | 1.5 | **0.1 - 200** | 30 |
| Test 4 | 236, 218, 201 | 1.5 | 30 | **10 - 60** |

simulations (Hartmann and Berry, 2017; van Diedenhoven et al., 2014). The second set of simulations consists of geometrical depth variations from 0.5 km up to 5 km, by fixing cloud top temperature at 203 K (Test 2a) and cloud bottom temperature at 237 K (Test 2b). The geometrical depth range follows similar values found by Sokol and Hartmann (2020). The third set of simulations increases IWP from 0.01 up to 200 g m$^{-2}$. These numbers follow realistic values for anvil clouds (Houze, 2014; Lawson et al., 2006, 2010; Sokol et al., 2024). Finally, Test 4 alters ice crystal effective radii in the Fu and Yi schemes. The Baran schemes have no direct dependence on ice crystal effective radius and are not included in this test. Additionally, Test 3 and Test 4 evaluate the IWP and effective radius sensitivities at three different cloud altitudes/temperatures. A summary of the inputs used here is shown in Table 2.

Profiles of ice mass mixing ratio $q_i$ are computed by fixing the IWP and calculating the uniform value of $q_i$ over the ice cloud depth that would generate that IWP (Figs. S3-S5 of the supplement). A fix IWP of 30 g m$^{-2}$ is used in Tests 1, 2 and 4. The IWP depends on $q_i$, air density $\rho_a$, and cloud thickness as follows:

$$IWP = \int_z q_i \rho_a \mathrm{d}z \qquad (2)$$

where $q_i \rho_a$ is the ice water content (IWC). $q_i$ spans an order of magnitude in our tests—from less than 0.01 g kg$^{-1}$ to $\sim$0.6 g kg$^{-1}$.

## 3 Results

### 3.1 How does cloud temperature influence CRH?

The effect of cloud temperature, or equivalently altitude, on CRH can be seen from Test 1 in the matrix visualizations of Fig. 2. These calculations use the Fu scheme, a fixed geometrical cloud depth of 1.5 km, IWP of 30 g m$^{-2}$, and ice crystal effective radius of 30 $\mu$m. The x-axis of each panel in Fig. 2 is the temperature of ice cloud formation, the y-axis is the corresponding

altitude of the cloud layer, and the color of each "pixel" in the matrix is the CRH, with red indicating heating and blue cooling. Each column in the matrix visualization is the vertical distribution of CRH obtained from one idealized single-column simulation run.

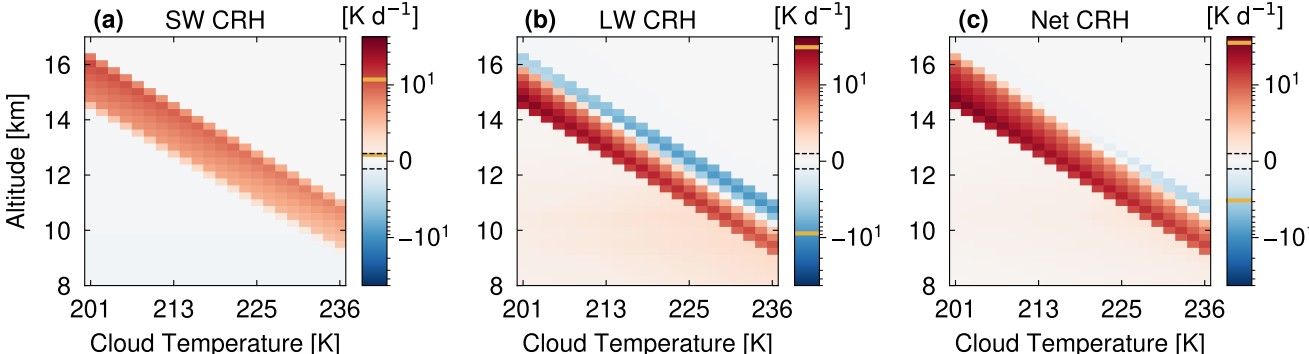

**Figure 2.** 22 CRH profiles for cloud temperatures from 201 K to 236 K computed by using the Fu scheme for SW (a), LW (b) and Net CRH (c). The cloud altitude is shown on the y-axis, while the heating rate magnitude (K d$^{-1}$) is represented in the colorbar. Yellow lines in the colorbar indicate the corresponding CRH range for each panel. Black dashed lines in the colorbar mark the limit between the linear and logarithmic scales (1 K d$^{-1}$).

Figure 2a shows how the SW component of CRH changes for different cloud temperatures. To start, the rightmost column of the matrix shows the CRH profile for a cloud at 236 K or about 10 km. The cloud warms throughout its depth, with larger heating on top due to stronger incoming SW fluxes. As we "transport" the ice cloud to higher positions in the troposphere (colder temperatures), the SW warming at the top of the cloud increases, reaching a maximum CRH of ∼12 K day$^{-1}$. This increase is due to the decreasing air density. The LW component of CRH exhibits a well-known "dipole" of in-cloud heating (10 K day$^{-1}$) and cloud-top cooling (-8.5 K day$^{-1}$, Fig. 2b). Reading the heating matrix visualization from right to left, we can see again how the heating increases with altitude until it dominates at the coldest cloud temperature of 201 K. While lower air density still explains the increase in CRH, strong temperature difference between a warm lower-tropospheric emission and cold ice cloud temperature results in maximum LW heating rates of ∼35 K day$^{-1}$ for this simulation. The LW component dominates net CRH for our chosen SZA (Fig. 2c). The CRH matrix visualizations for the other schemes are shown in Fig. S6 of the supplement.

We turn next to how these CRH profiles change with ice optical scheme. While the heating rate matrices of Fig. 2 shows absolute CRH, those of Fig. 3 now show interscheme CRH differences. The simplest Fu scheme has the strongest SW heating; the more complicated schemes all produce weaker SW heating—and hence a negative difference—relative to the Fu scheme (Figs. 3a, 3d, and 3g). The Baran14 scheme has a peak difference from Fu of -1 K day$^{-1}$ (-18%) at the warmest subzero temperatures, and the difference weakens with height (Fig. 3a).

These SW CRH differences can be explained mainly by inter-scheme differences in the shortwave absorption coefficient $K_{abs}^{SW}$ (Fig. 1 and Fig. 4i). The SW CRH in Yi13 is 18% less than in the Fu scheme at the top layer of the high altitude cloud.

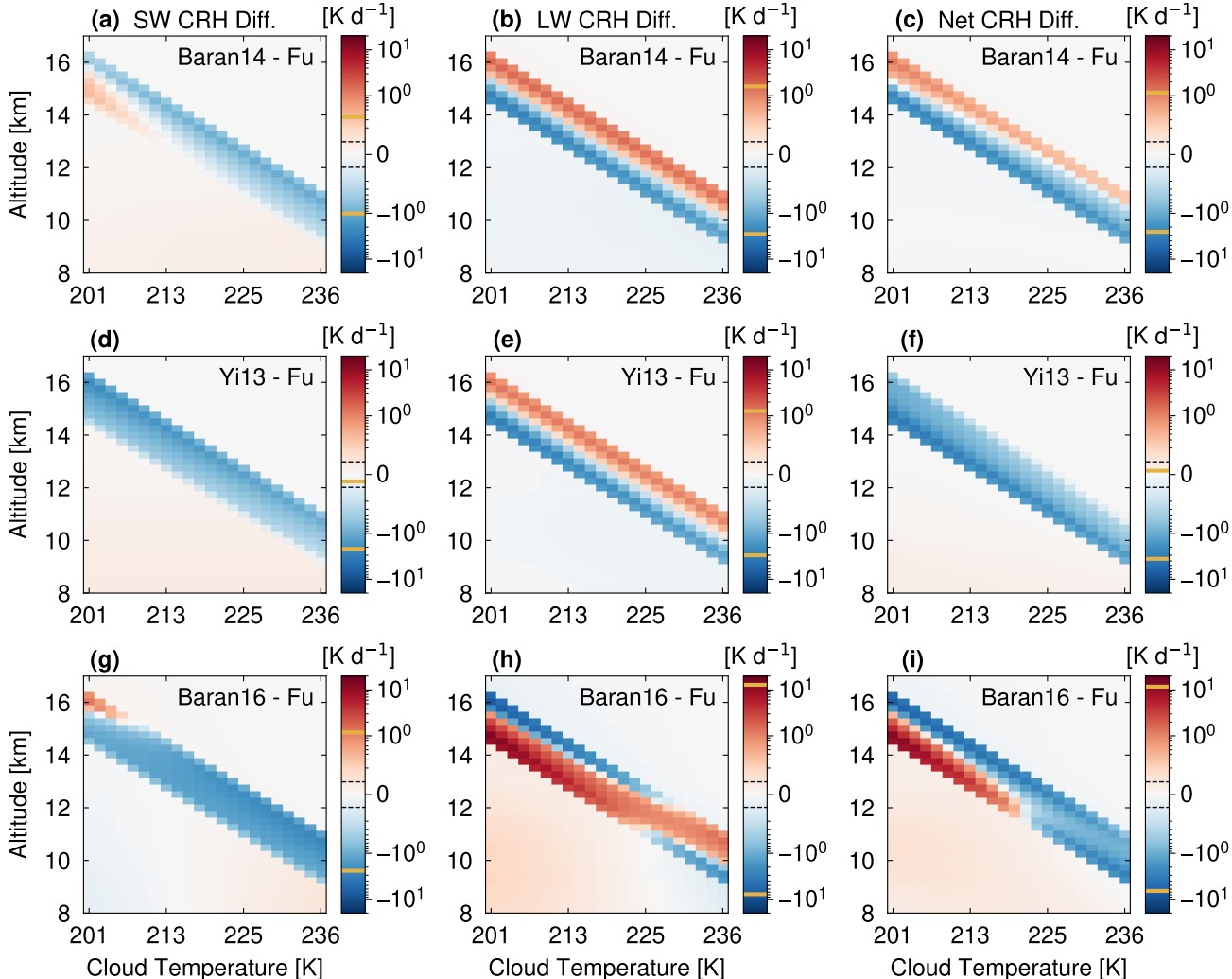

**Figure 3.** Heating rate matrix visualizations, including 22 CRH difference profiles each for cloud temperatures from 201 to 236 K in Test 1. The left column shows SW CRH differences, the middle column LW CRH differences, and the right column net CRH differences. Rows are organized by scheme from least to most complex, top to bottom: Baran14-Fu (top row), Yi13-Fu (middle row), and Baran16-Fu (bottom row). The cloud altitude is given on the y-axis, while the heating rate difference magnitude ($\Delta$CRH in K day$^{-1}$) is given logarithmically in the colorbar. The limit between the linear and logarithmic scale is 0.1 K day$^{-1}$. As in Fig 2, yellow lines in the color bar indicate the corresponding $\Delta$CRH range for each panel.

At low altitudes, the SW CRH from Baran16 is 42% less than that from Fu; however, by the highest altitudes, it is 10% higher. $K_{abs}^{SW}$ can again explain this trend (Fig. 4k): A gradient exists in $K_{abs}^{SW}$ with lower values in Baran16 at low altitudes and higher values in Baran16 at high altitudes.

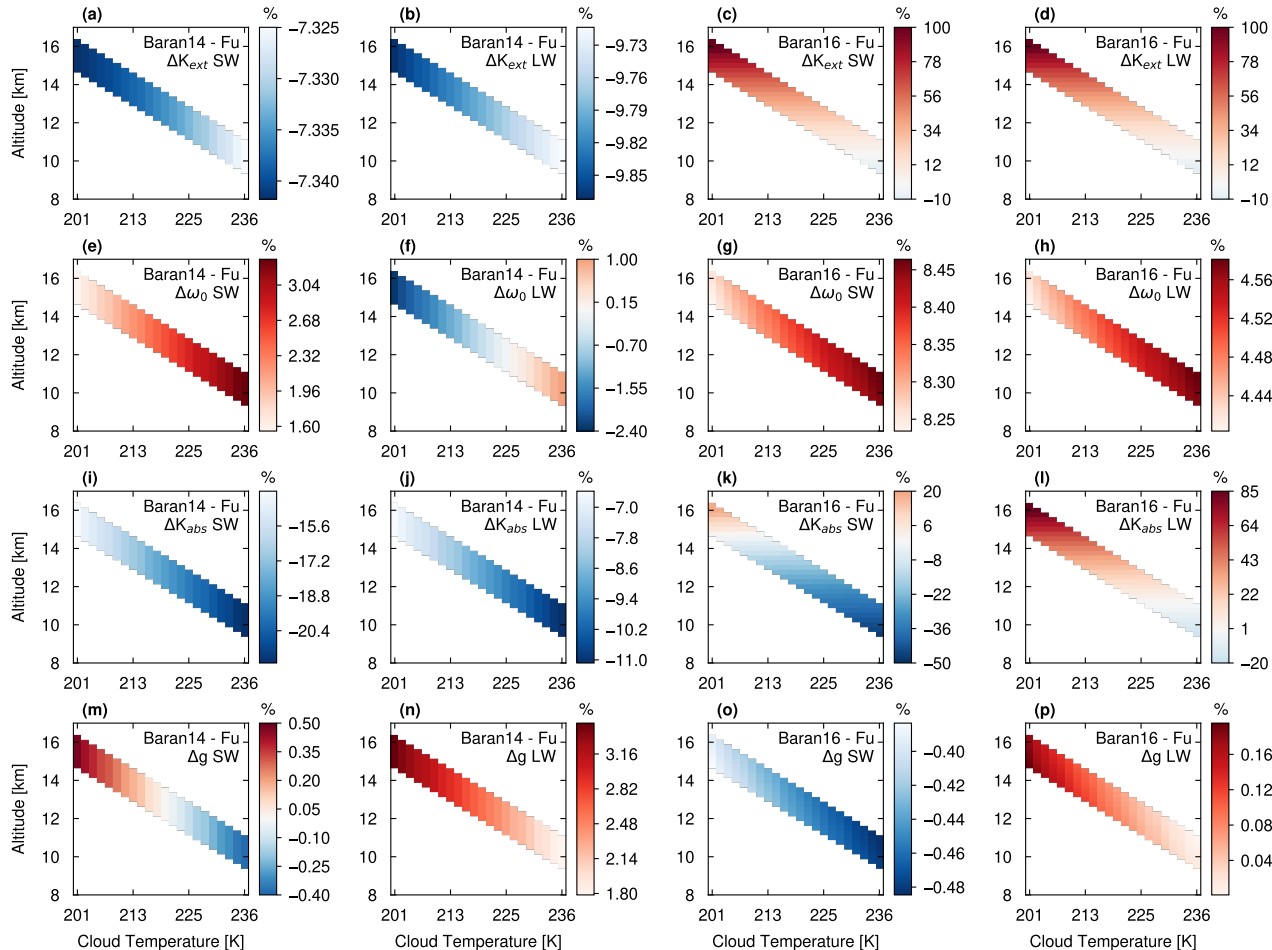

**Figure 4.** Matrix visualizations including 22 interscheme relative difference profiles for the $K_{ext}$, $\omega_0$, $K_{abs}$ and $g$ (top to bottom rows), each for mid-cloud temperatures from 201 K to 236 K (Test 1). First and second column shows Baran14-Fu interscheme difference for SW and LW components respectively. The same components are shown in the third and fourth column for Baran16-Fu interscheme comparison.

For Baran14, we can understand the SW CRH differences in terms of $K_{abs}^{SW}$ but also $g^{SW}$ (Fig. 4m). As is mentioned in Sect. 2.1, $g$ quantifies the amount of forward versus backward scattered radiation by ice crystals. $K_{abs}^{SW}$ in Baran14 is between 14 and 22% smaller that in the Fu scheme, while $g^{SW}$ changes from 0.4% smaller to 0.5% larger with increasing altitude. For our given SZA, a larger $g$ increases the chances of absorption at the bottom of the cloud layer, likely driving the small positive deviation at the uppermost levels.

Larger magnitude differences exist for the LW CRH. Both the Baran14 and Yi13 schemes produce weaker in-cloud heating and cloud-top cooling than the Fu scheme (Figs. 3b and 3e). These differences in the LW CRH increase with altitude for these schemes. At the highest altitudes, both Baran14 and Yi13 have roughly 10% less in-cloud heating and 50 to 60% less cloud-top

cooling than the Fu scheme. As for the SW CRH, these differences can be explained in terms of weaker absorption in these schemes: $K_{abs}^{LW}$ in Baran14 is roughly 10% smaller than that in Fu (Fig. 4j) and in Yi13 roughly 7.5% smaller (Fig. 1).

Finally, an interesting inversion in LW CRH differences exists for the Baran16 scheme (Fig. 3h). At low levels, it follows the behavior of Baran14 and Yi13 with weaker in-cloud heating and cloud-top cooling than the Fu scheme. At upper levels close to the TTL, however, temperature dependence in the Baran16 scheme leads to opposite behavior, in which in-cloud heating is 36% larger and cloud-top cooling is almost 3 times larger than in the Fu scheme. A strong gradient in both $K_{ext}^{LW}$ and $K_{abs}^{LW}$ with temperature explains this inversion (Figs. 4d and 4l). In particular, $K_{abs}^{LW}$ is 85% higher in Baran16 than in Fu at the coldest temperatures. Differences in $g^{LW}$ are much smaller in this case, indicating that a temperature-dependent $K_{abs}^{LW}$ has a stronger effect than a non-temperature dependent ice crystal habit description.

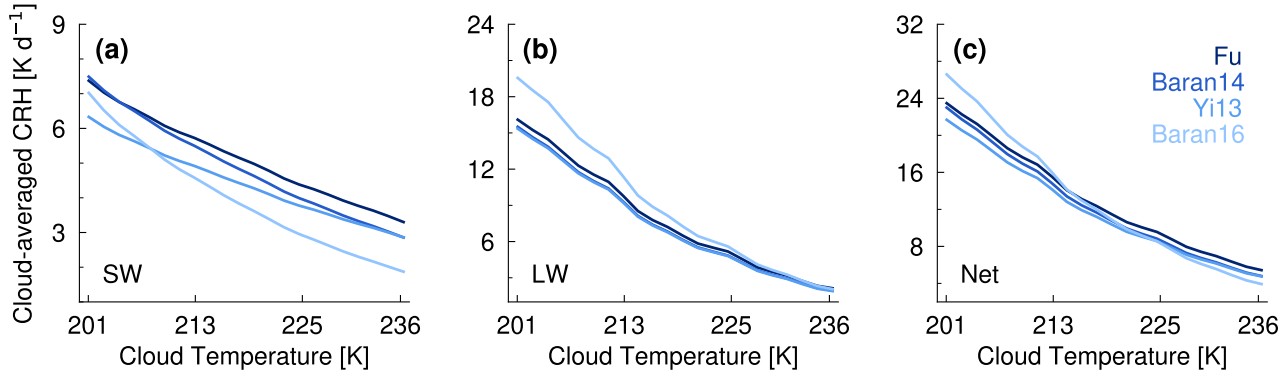

**Figure 5.** SW (a), LW (b) and Net (c) Cloud-average CRH (y-axis) evaluated for each single-column profile shown in Fig. 2, across cloud temperatures from 201 to 236 K in Test 1 (x-axis). All four ice optical schemes under test are shown from darker blue to light blue, indicating low to high ice complexity level.

To summarize the overall atmospheric impact of each optical scheme, Figure 5 shows the cloud-average radiative heating rate ($\overline{CRH}$), computed as:

$$\overline{CRH} = \frac{\int_z CRH\, q_i \rho_a \mathrm{d}z}{\int_z q_i \rho_a \mathrm{d}z} \tag{3}$$

Optical schemes such as Baran14 and Yi13, both result in weaker SW $\overline{CRH}$ relative to Fu, following the trend discussed above. However, no difference in Baran14 SW $\overline{CRH}$ is found at cold clouds, showing the limitations when analyzing a not vertically-resolved CRH interscheme difference. Equally stronger in-cloud and weaker cloud-top heating results in net zero vertically integrated SW $\Delta\overline{CRH}$. Both Baran14 and Yi13 show a uniform weakening of 5% in LW $\overline{CRH}$ throughout all temperature ranges. However, it is not possible to detect the inversion in Baran16 LW CRH mentioned in the previous paragraph, with Baran16 LW $\overline{CRH}$ always being higher than the other schemes. In contrast, Baran16 Net $\overline{CRH}$ changes from lower to higher values than the other schemes when the cloud temperature increases, mainly due to the contribution of the SW component. While some features of the CRH interscheme difference can be detected in an average number, which in turn can be useful

 for satellite comparison at the top of the atmosphere (TOA), the vertically resolved CRH difference is a key component when evaluating other atmopsheric mechanisms, as radiative induced vertical motion. Radiative-related cloud processes cannot be completely understood with only a cloud-average radiative heating rate analysis.

## 3.2 How does cloud depth influence CRH?

We turn next to results of Test 2, which studies sensitivity of CRH to cloud geometrical depth across optical schemes (Table 2). First, in Test 2a, increasing cloud depth when keeping cloud top temperature fixed weakens the CRH at a given altitude (Fig. S7 of the supplement). Since we fix IWP and infer $q_i$ across the cloud depth (see Eq. (2) and Fig. S4a), there is less ice condensate at a given level as the ice cloud deepens. Less condensate then means less SW absorption at a given level. A similar trend is found in the LW CRH matrices from all schemes: As the ice cloud depth increases, less condensate absorbs and reemits at a given level, weakening both in-cloud heating and cloud-top cooling.

Interscheme differences in Test 2a mirror those of Test 1 with weaker SW and LW absorption from the Yi13 and Baran14 schemes relative to the Fu scheme (Fig. 6). The exception is in thin clouds at high altitudes, where both Baran schemes show a stronger SW heating (Figs. 6a and 6g). In Baran14, the $K_{abs}^{SW}$ is lower than in Fu (Fig. S8i). The stronger SW heating at high levels must then be due to the scattering parameters. $\omega_0^{SW}$ is smaller and $g^{SW}$ is larger in Baran14 (Figs. S8e and S8m), meaning that this scheme predicts relatively more absorption and forward scattering than Fu. Both of these behaviors promote in-cloud absorption and the positive SW CRH anomaly at high altitudes.

For Baran16, the temperature-dependent $K_{abs}^{SW}$ causes stronger absorption than in Fu, at the top of thin clouds. This property difference is the source of the largest magnitude SW CRH differences. As in Baran14, there is a gradient of $g^{SW}$ through the geometrical depth range, as a result of the change of $q_i$ with depth (Fig. S8o). $g^{SW}$ in Baran16 reaches similar values than Fu for the geometrically thin cloud, meaning that SW absorption is the main driver of the SW CRH anomaly here. However, less forward scattering in geometrically thick clouds leads to weaker SW CRH at the top layers. The SW CRH differences are smallest between Yi13 and Fu, due to a smaller $K_{abs}^{SW}$ in the Yi13 scheme (Fig. 1).

As for Test 1, larger differences are present in the LW component across the schemes. Baran16 has 20% more in-cloud heating and 130% more cloud-top cooling at the uppermost levels relative to Fu. These CRH differences "dilute" as we deepen the cloud from left to right in the heating rate matrices. In contrast to the SW component, the LW CRH differences in Baran14 do not look like those in Baran16. Baran16 has dramatically more upper-level LW CRH than Fu (+10 K day$^{-1}$), whereas Baran14 has slightly less. $K_{abs}^{LW}$ is again the property driving these differences. $K_{abs}^{LW}$ is 70% larger in Baran16 than in Fu for cloud temperatures colder than 210 K (Fig. S8l); in Baran14, the $K_{abs}^{LW}$ is 7% smaller at the highest altitudes (Fig. S8j). LW CRH difference dominates at cloud bottom, while the SW CRH difference dominates toward the top of cloud for net CRH differences.

In Test 2b, we fix the cloud bottom temperature and decrease the cloud top temperature (Fig. 7). These simulations also fix IWP, so that $q_i$ at a given level decreases as cloud depth increases (Fig. S4b). Indeed, the absolute heating rate matrix for Test 2b shows enhanced heating in the less deep clouds where condensate is more concentrated (Fig. S10). However, the difference in CRH across cloud depths is more pronounced in Test 2a than in Test 2b because $q_i$ must be greater at high altitudes than at

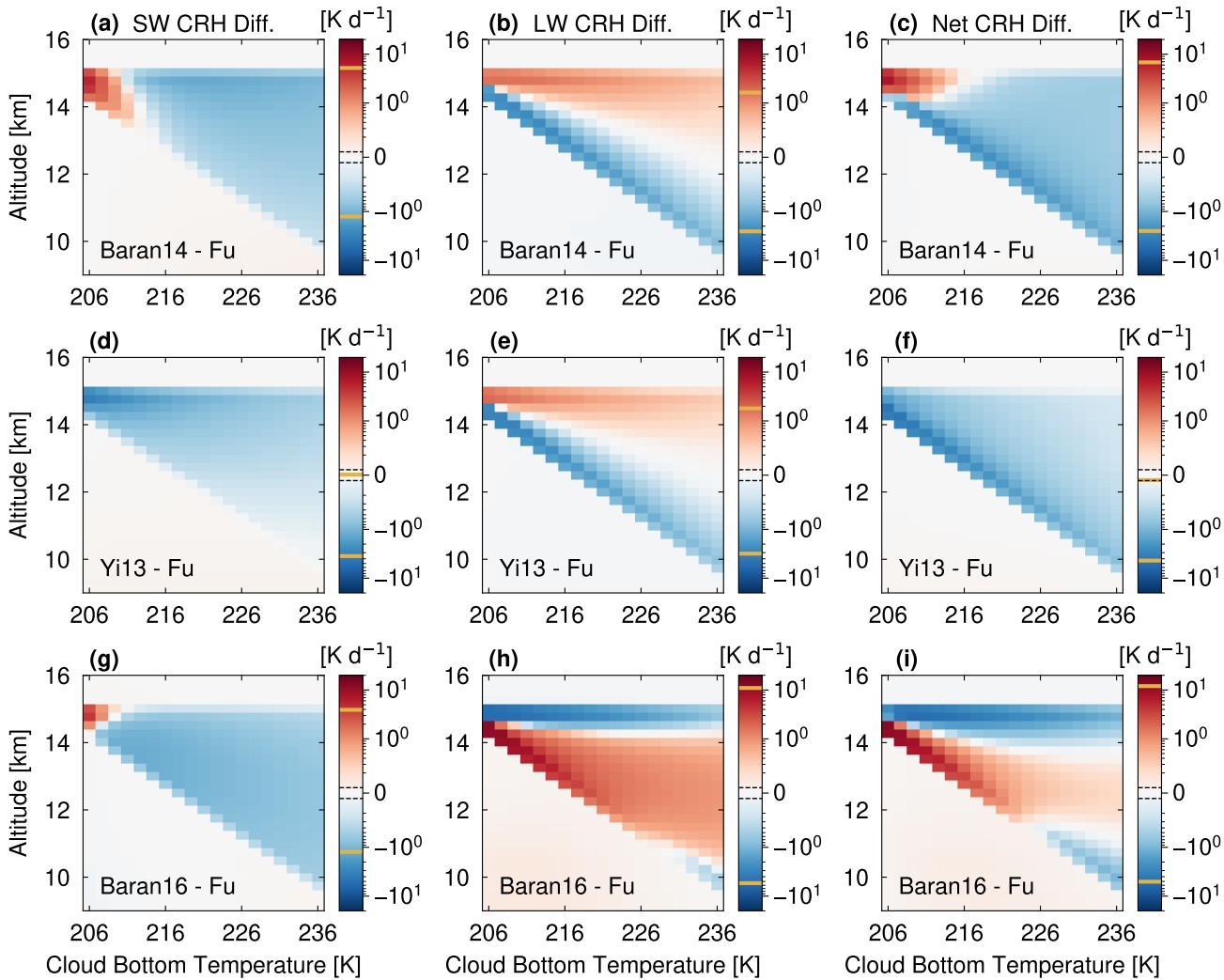

**Figure 6.** Heating rate matrix visualizations, including 19 CRH difference profiles each for cloud depths from 0.5 to 5 km, with fixed cloud top temperature of 203 K (Test 2a). Panels, axes, and colorbar are as in Fig. 3.

low altitudes to generate a given IWP for a fixed geometric depth. Differences in absorption are therefore larger in Test 2a with a geometrically thin cloud at high altitudes than in Test 2b with a geometrically thin cloud at lower altitudes.

Despite these quantitative differences between Test 2a and Test 2b, we see qualitatively similar behavior in SW and LW CRH across the schemes. Including more ice crystal habits in Baran14 and Yi13 leads to 12% more and 14% less SW heating, respectively. As mentioned above, $g^{SW}$ is larger and $K_{abs}^{SW}$ is only slightly lower in Baran14 than Fu, so that there is a small positive difference at low levels for the most dense clouds (Fig. 7a). The Yi13 scheme has minimal $g^{SW}$ differences for the

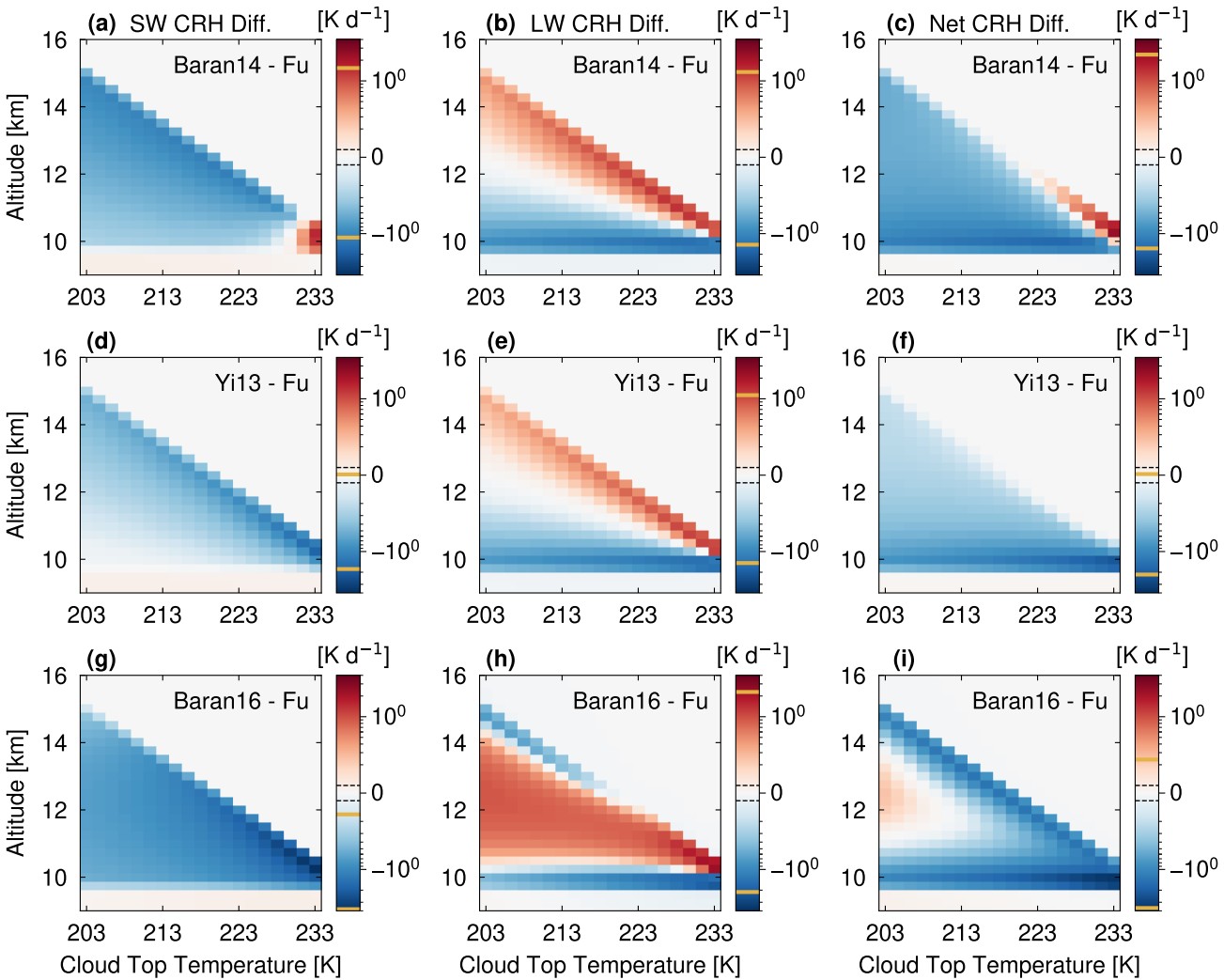

**Figure 7.** Heating rate matrix visualizations, with the same depth range as Fig. 6 and with a fixed cloud bottom temperature of 237 K (Test 2b). Panels and axes are as in Fig. 3. The colorbar ranges from -4 to 4 K day$^{-1}$. The limit between the linear and logarithmic scale is 0.1 K day$^{-1}$.

effective radius used here and its smaller $K_{abs}^{SW}$ produces only negative SW CRH differences. LW differences are qualitatively the same but of smaller magnitude between Test 2a and Test 2b.

  In contrast, inclusion of temperature dependence in Baran16 results in a different trend for Test 2b than in Test 2a. The SW CRH is weaker in Baran16 than in Fu with about 30% less heating in the thinnest clouds and 13% less in the thickest ones due to a weaker $K_{abs}^{SW}$ below 210 K in Baran16 (Fig. S11k). LW CRH differences between Baran16 and Fu also show a unique

structure (Fig. 7h) across cloud depths, Baran16 predicts less LW absorption at the warmest temperatures. For the clouds of

greater depth, it also predict roughly 50% more in-cloud heating and 10% more cloud-top cooling than the Fu scheme. The prominent vertical gradient in $K_{abs}^{LW}$ for Baran16 drives these differences (Fig. S11l): It varies from 10% smaller at warmer temperatures to 63% larger at colder temperatures than $K_{abs}^{LW}$ in the Fu scheme, enhancing the absoprtion capacity in the upper half of the ice clouds. Again in contrast to Test 2a, the Baran16-Fu interscheme difference in SW CRH dominates the Net CRH for thin clouds. Similar to Fig. 5, the $\overline{CRH}$ for Test 2a and 2b are shown in Fig. S9 and S12 of the supplement, respectively.

### 3.3  How does IWP influence CRH?

We study the CRH sensitivity to IWP in Test 3. We fix the geometric depth and altitude of the ice cloud and instead vary the amount of condensate within it (Table 2). The highest IWP of 200 g m$^{-2}$ corresponds to $q_i$ of 0.33 g kg$^{-1}$ for the lowest cloud altitude and qi of 0.66 g kg$^{-1}$ for the highest cloud altitude (Fig. S5 of the supplement).

As expected, CRH increases with IWP and, as in Test 1, with altitude (Fig. S13). The SW CRH increases from roughly 0.02 to 20 K day$^{-1}$ for the lowest cloud altitude. This change is twice as large for the highest cloud altitude. While the in-cloud LW heating and cloud-top LW cooling dipole is always present in previous tests, all four schemes show an in-cloud LW heating for IWP from 4 to 20 g m$^{-2}$. The LW CRH structure dominates the net CRH, but the SW cloud top heating can surpass the LW cloud-top cooling in some cases.

We then visualize CRH interscheme differences for this test (Fig. 8). In contrast to Test 1, we see larger differences in the SW CRH than the LW CRH for Baran14 and Yi13. For both schemes, we see interesting inversions in the SW CRH differences at high IWPs. For example, in Baran14, SW CRH changes from 50% smaller to 30% larger at an IWP of roughly 30 g m$^{-2}$. A similar change occurs in $g^{SW}$ (Fig. S14m). A smaller $g^{SW}$ in Baran14 at low IWPs enhances backwards scattering of SW radiation at the cloud top relative to Fu. Conversely, a slightly higher $K_{abs}^{SW}$ and larger $g^{SW}$ in Baran14 at large IWPs enhances forward scattering of SW radiation into the cloud, where it can be absorbed. The inversion in Yi13 differences is much more muted (Fig. 8d). As in preceding tests, the weaker $K_{abs}^{SW}$ in Yi13 relative to Fu drives its weaker SW CRH. The negative difference in $K_{abs}^{SW}$ ($\sim$-20 %) outweighs the positive difference in $g^{SW}$ ($\sim$2%) in Yi13, except at IWP greater than 100 g m$^{-2}$, where the higher forward scattering relative to Fu enhances the chances of SW radiation to be absorbed at the bottom of the cloud layer.

Interscheme differences in LW CRH are as in Tests 1 and 2 with weaker in-cloud heating and cloud-top cooling in Baran14 and Yi13 than in Fu (Figs. 8b and 8e). Importantly, this trend is maintained below an IWP of 10 g m$^{-2}$, although, with a different CRH vertical structure. For ice clouds with the lowest amounts of condensate, the cloud-top cooling becomes less important, and LW CRH differences are negative throughout the cloud depth. As IWP increases, the difference in $\omega_0$ goes from positive to negative between Baran14 and Fu (Fig. S14f), resulting in a slightly higher $K_{abs}^{LW}$ in Baran14 relative to Fu. This increased absorption relative to scattering drives the positive anomaly in LW CRH at high IWP. The LW CRH interscheme difference dominates the Net CRH anomalies in the bottom layer of the clouds, while the SW CRH differences is important in the top layers.

As in Test 2, the Baran16 scheme shows contrasting behavior to Baran14 and Yi with more in-cloud heating and cloud-top cooling than Fu at high altitudes (Fig. 8h). The absolute differences are especially pronounced at high IWP with 20% more

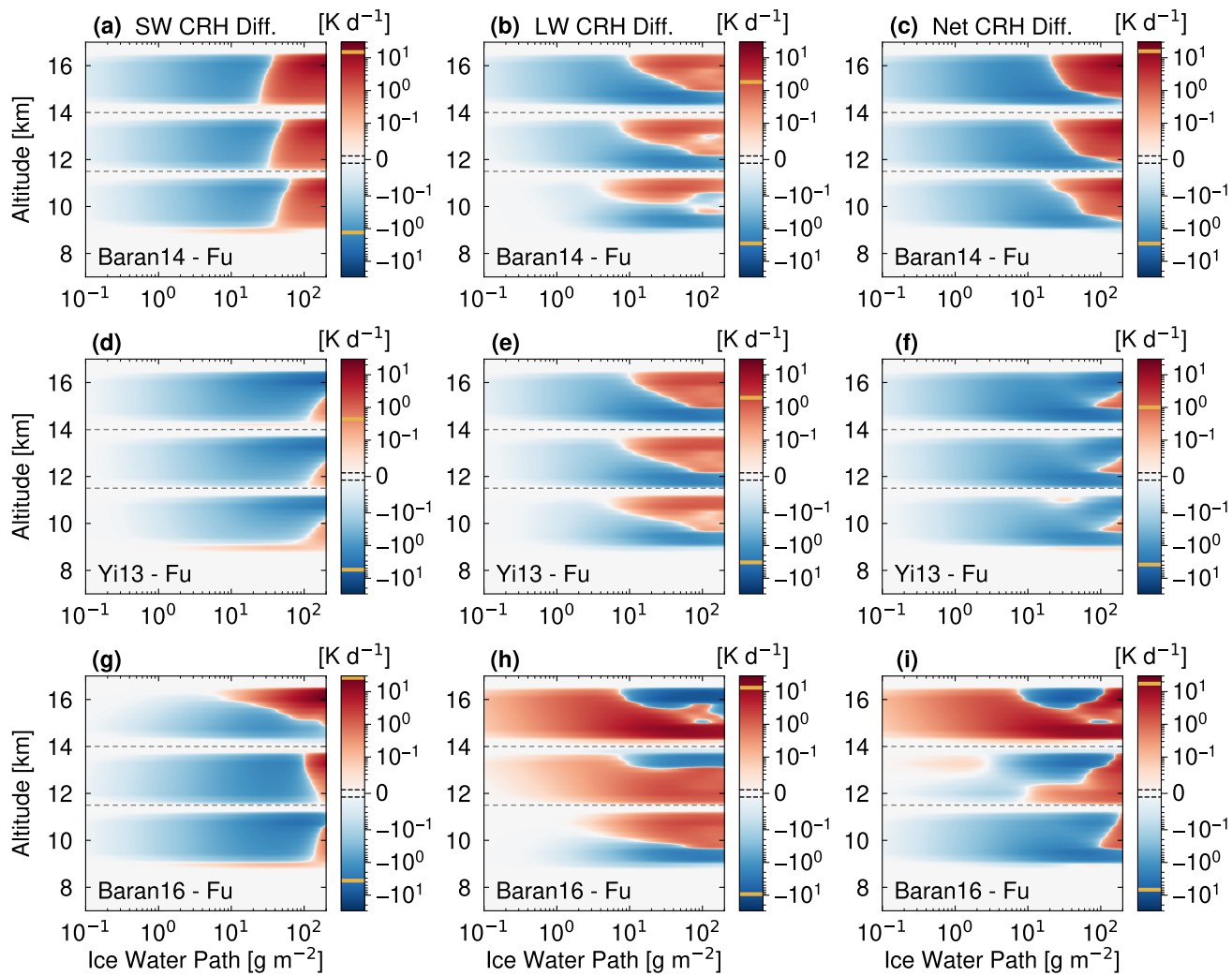

**Figure 8.** Heating rate matrix visualizations, including CRH difference profiles for IWPs from 0.1 to 200 g m$^{-2}$ in Test 3. Three separate differences are shown for cloud temperatures of 201 K, 218 K, and 236 K, corresponding to high, middle, and low altitudes. Panels, axes, and colorbar are as in Fig. 3. The limit between the linear and logarithmic scale is 0.01 K day$^{-1}$.

in-cloud heating and 50% more cloud-top cooling for the cloud layer at the uppermost altitude. Temperature dependence in $K_{abs}^{LW}$ is key here, with the absorption coefficient almost two times as large in Baran16 as Fu at the coldest temperatures and across IWP values (Fig. S14l). The $\omega_0$ difference does not change from positive to negative in Baran16, indicating that the absorption coefficient trends dominate.

### 3.4 How does ice crystal effective radius influence CRH?

We finish by examining the sensitivity of CRH to ice crystal effective radius, $r_{\text{eff}}$, in Test 4. Only the Fu and Yi13 scheme parameterize ice optical properties as a function of $r_{\text{eff}}$ (Fig. 1, so the Baran schemes are not included in Test 4. The CRH profiles computed for effective radii ranging from 10 to 60 $\mu$m are shown in Fig. S15, with a fixed cloud depth of 1.5 km and fixed IWP of 30 g m$^{-2}$. Three independent RT calculations are included for each of three cloud temperatures, similar to Test 3 (Table 2). The SW CRH in the Fu scheme reaches 20 K day$^{-1}$ at the top layers of colder clouds composed of small crystals (Fig. S15a). This SW CRH weakens for warmer clouds, as in previous tests, and for larger $r_{\text{eff}}$. Smaller crystals scatter more than larger ones and also favor forward scattering (Fig. S1d), both of which promote opportunities for SW absorption within the cloud.

A similar trend is found in LW CRH for the Fu scheme (Fig. S15b), with smaller crystals producing higher CRH rates. The in-cloud heating and cloud-top cooling peak at 60 and 20 K day$^{-1}$, respectively, for ice crystals of $r_{\text{eff}}$ less than 20 $\mu$m in colder clouds. As $r_{\text{eff}}$ increases, the LW in-cloud heating decreases to 20 K day$^{-1}$, and the cloud-top cooling decreases to 4 K day$^{-1}$ at high altitudes. While the heating-cooling dipole is still present in the net CRH for small crystals (Fig. S15c), a net heating dominates for colder clouds with ice crystals of $r_{\text{eff}}$ larger than 20 $\mu$m.

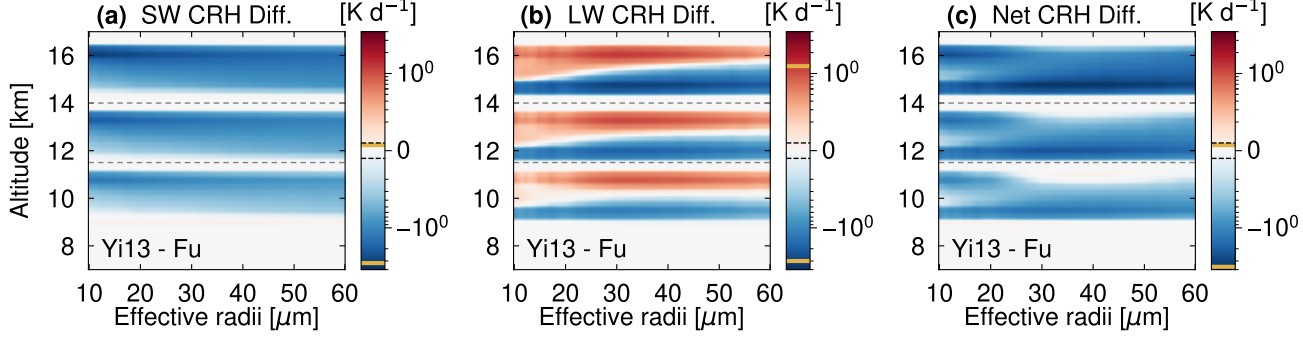

**Figure 9.** Heating rate matrix visualizations including CRH interscheme difference profiles for ice crystal effective radii ranging from 10 to 60 $\mu$m (Test 4). Three separate profiles are calculated for low, middle, and high cloud temperatures of 201 K, 218 K, and 236 K. Columns, axes, and colorbar are as in Fig. 8.

Echoing previous tests, the weaker $K_{abs}^{SW}$ and slightly higher $g^{SW}$ from more complex crystals in Yi13 cause weaker SW CRH than in Fu, across the full range of effective radii. The largest SW CRH differences are in the coldest cloud layer. The SW CRH interscheme difference at the top layer of these coldest clouds changes from -3 K d$^{-1}$ to -1.5 K d$^{-1}$ as $r_{\text{eff}}$ increases (Fig. 9a). For LW CRH, Yi13 predicts weaker in-cloud LW heating and cloud-top LW cooling for all effective radii relative to Fu. The LW CRH differences are highest for ice crystal radii between 25 and 40 $\mu$m. The lower $K_{abs}^{LW}$ together with a larger $g^{LW}$ explain these LW CRH anomalies (Figs. S2i and S2l). Finally, a weaker net heating dominates in Yi13 at all altitudes and ice crystal effective radii, with a -10% difference at the bottom of the coldest cloud.

## 4   Discussion and conclusions

Ice crystals in high clouds can be found with varying degrees of complexity, impacting the bulk optical properties of an ice cloud. Different ice optical schemes have been proposed assuming different ice crystal habit distributions, either based on in-situ measurements or optical particle model calculations. Here, we run the ecRad radiative transfer scheme with four such ice optical schemes over an atmospherically relevant parameter space for ice clouds to understand their effect on CRH. We propose heating rate matrix visualizations to present our results across the parameter space in a condensed format. We consider two main characteristics that distinguish each scheme: first, the ice crystal habits and complexity assumed and second, the variables on which the parameterized optical properties depend, including ice crystal size, ice mass mixing ratio, or temperature.

In the current analysis, we consider the Fu scheme as a starting point in terms of ice crystal complexity, as this scheme bases its calculations on smooth hexagonal columns, and depends only on the size of the ice crystals. Schemes such as Baran14, Baran16 and Yi13 increase the ice crystal habit description. The Baran schemes use a simpler ensemble of ice crystal habits, including surface roughness in the SW calculations, and Yi13 a more elaborate one, including hollowness and surface roughness properties in both SW and LW parameterizations. In contrast to the other schemes, Baran16 includes temperature as an input in the parameterization. Interscheme differences in CRH then indicate which feature of the schemes is relevant under which atmospheric conditions. Moreover, we find that most of the SW / LW CRH differences can be explained in terms of differences in the absorption coefficient and, secondarily, by those in the asymmetry parameter.

A key result is that ice complexity weakens heating rates for both SW and LW components, especially in geometrically thin clouds at high altitudes. The difference is also particularly pronounced for clouds with high IWP, resulting in net CRH differences of -2.5 K d$^{-1}$ at cloud bottom in $r_{\mathrm{eff}}$-dependent schemes and 15 K d$^{-1}$ at cloud top in $q_i$-dependent schemes, when compared with a less sophisticated scheme. In general, when more ice crystal habits are included, the absorption capacity of ice clouds decreases greatly, while the forward scattering increases to a lesser extent, resulting in all weaker SW CRH, LW in-cloud heating, and cloud-top cooling.

The sensitivity test presented here looks to show the impacts of ice optics assumptions under different cloud scenarios based on realistic ranges. However, existing evaluations result on similar trends to those derived in this test. Previous works shows a general trend of decreasing downward SW flux at the surface and increasing upward SW flux at the TOA when using aggregate ice crystals relative to hexagonal columns (Yang et al., 2012). Higher single-scattering albedo from the aggregate crystals are the primary explanation for these trends. These effects translate to a decrease in SW CRE at TOA of roughly 1 W m$^{-2}$ (Järvinen et al., 2018). Our results follow a similar trend where schemes such as Baran14 and Yi weaken the LW cooling rate at the cloud top due to less LW emission, and as a result of generally less absorption capacity by the ice cloud under test. Moreover, previous studies show both weaker SW and LW CRH when switching to a more complex ice scheme, whether studying CRH in an extratropical cyclone Keshtgar et al. (2024) or in global scale simulations Zhao et al. (2018).

The inputs used in the optical scheme can have a strong influence on radiative output and CRH profiles. For certain ice clouds, temperature dependence has a larger impact on optical property differences than does inclusion of complex habits or surface roughness. In this case, the Baran16 scheme shows small weaker heating rates for geometrically thin clouds at low

altitudes, as a possible consequence of the inclusion of more habits apart from only hexagonal columns. However, net heating rates at the bottom layer of geometrically thin clouds at high altitudes can be as high as 12 K d$^{-1}$ relative to a size-dependent and simpler complexity scheme. This is due mainly to the $T^{-4}$ temperature dependence of the bulk mass absorption coefficient in Baran16, which has a bigger impact than the $q_i$ parameter, and allows a relatively strong SW and LW cloud top-bottom heating respectively.

These results can be helpful from a practical standpoint, by evaluating the impact of switching ice optical schemes in current RTM and NWP models. Either weaker CRH due to ice complexity or stronger CRH due to temperature-dependent ice optical schemes can have important impacts across scales. When evaluating the evolution of convective systems and individual cloud scale processes, different magnitudes of calculated net CRH would alter the lifetime of anvil clouds (Sokol and Hartmann, 2020; Hartmann and Berry, 2017; Gallagher et al., 2012). At a larger scale, less heating in the tropical upper troposphere due to inclusion of more complex ice crystals would weaken tropical overturning by reducing the latitudinal temperature gradient at high altitudes (Gasparini et al., 2023). In similar ways, changes in calculated CRH due to assumptions of ice crystal habits, would impact circulation pattern estimations, by changing for example, the magnitude of the poleward jet shifting (Voigt and Shaw, 2015).

While the ice optical schemes tested here are based on inputs and ice cloud properties relevant for tropical regions, it is highly recommended to pre-evaluate the right scheme to use in current climate and forecast simulation, depending on the region-time and meteorological context under evaluation (Noel et al., 2006; van Diedenhoven, 2018; Sato and Okamoto, 2023). Considering that ice crystal size can also change in space and time (Pasquier et al., 2023; Llombart et al., 2020), in-situ measurements of region-specific behaviour in ice complexity and sizes are still necessary. From other side, when including temperature in the retrieval of optical schemes, we have the advantage of using a prognostic variable that can be retrieved with less uncertainty than ice crystal size. Temperature and ice concentration-dependent schemes can behave relatively similar under certain conditions, however, radiative analysis in specific cases of high IWP in cold temperature environments has to be treated cautiously.

The idealized single-column tests used in this work assume a vertical uniform distribution of $q_i$ and $r_{eff}$ through the cloud layers, which is not a typical realistic case. However, the perturbation method allows us to focus on independent impacts of $q_i$ and $r_{eff}$ on the CRH profiles in the tropics. From an atmospheric modeler point of view or for microphysics studies, the sensitivity tests shown here offer a hint on which parameter evaluated in this work is causing the biggest impact on computed CRH. Cases where a thin high peak of the $q_i$ profile located at cold temperatures result in the largest differences of the retrieved CRH by the variety of ice complexity assumption. Therefore, these cases should be analyzed and processed with caution when are present in forecast and climate models. Moreover, while we expect SZA to result on a variety of SW CRH structure, as it changes the chances of radiation to be absorbed when SZA is high, other parameters assumed as fixed in this test, such as SW albedo and LW emissivity, would have a systematic impact on CRH profiles, and therefore on the interscheme difference trend.

As discussed previously, preliminary work including more realistic contexts, have already shown similar qualitative trends on CRH when ice complexity is included. However, more evaluations, by using storm-resolving models for instance, under specific cloud structures and characteristics will help to assess the utility and performing of the sensitivity test presented here. Including remote sensing products in future analysis is also a necessary part of the evaluation workflow, but caution must be

considered as even satellite products assume ice crystal characteristics when retrieving CRH, $q_i$ or $r_{eff}$ profiles (Wu et al., 365  2024; Ren et al., 2023; Delanoë and Hogan, 2010).

*Code and data availability.*  ecRad radiative transfer model (RTM), version 1.5.0 is available in https://github.com/ecmwf-ifs/ecrad (Hogan and Bozzo, 2018). CRH calculations and codes for figures are available in zenodo online repository https://doi.org/10.5281/zenodo.13932540 (Sepulveda Araya, 2024a) and https://github.com/EdgardoSepulveda/ice-crh-1column. Data inputs, configuration files for simulations, simulation outputs, and CRH tables are available in zenodo online repository https://doi.org/10.5281/zenodo.13932430 (Sepulveda Araya, 2024b).

*Author contributions.*  EISA run the simulations and coding. SCS and AV developed the matrix visualization idea. All authors reviewed the writing.

*Competing interests.*  The authors declare that they have no conflict of interest.

*Acknowledgements.*  This research was supported by start-up funds through the University of Arizona (UA), as well as UA Research, Impact, and Innovation International Research Grant No. 2100255.

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
