# Peer review of "Ice crystal complexity leads to weaker ice cloud radiative heating in idealized single-column simulations"

_EGUsphere, 2024_

## Referee Comment (RC1)

This paper by Sepulveda Araya et al compares ice-cloud radiative heating rates derived from four different radiative transfer schemes. The paper is very well motivated, well organized, and well written. The analysis is sufficiently thorough to explain why the different schemes produce different heating rates, and the presentation and discussion are very clear. The results of the intercomparison are quite nuanced, but there are two primary points: (1) accounting for ice crystal complexity generally weakens CRH, and (2) temperature-dependent schemes can lead to large CRH differences at cold temperatures characteristic of the upper troposphere.

The paper is well suited for ACP and will appeal not only to readers in the ice microphysics, cloud-radiation, and cloud-climate communities. I learned a lot from the paper, and think it has several valuable lessons for folks such as myself, who care about the role of ice clouds in climate but have little technical knowledge of the radiative transfer schemes on which our work relies.

The paper could be published as is, with some technical corrections. Nevertheless, I had a couple of general suggestions and number of line comments (mostly just clarification), so I will suggest minor revisions in case the authors wish to incorporate any of it. I look forward to seeing the paper published.

-Adam Sokol

**General comments/suggestions** (all purely optional)

1. The figures and analysis all focus on vertically resolved CRH which, for reasons discussed in the Intro, is important for many reasons. Layer-averaged CRH may provide an additional, simpler perspective on interscheme differences. In Fig 2b,e, for example, showing layer-averaged CRH would indicate whether the positive and negative lobes of the dipole cancel out, or whether the net effect is nonzero. This could help qualify which downstream impacts may be affected by interscheme differences; for example, a layer-averaged difference near 0 wouldn't be so important for planetary-scale circulation, but ~10 K/day certainly would be.

   For Fig S8 showing the layer-averaged CRH would convey whether the total heating is dependent on cloud thickness, or if the total is invariant but distributed differently throughout the cloud as the thickness varies. If the schemes produced different results in this regard, I would think that would be quite interesting.

   Another option (closely related to the layer-averaged heating) would be to show the TOA CRE on top of each heating matrix. The CRH differences suggest that the TOA CRE differences would be very large in some of these cases.

2. The assumption of vertically uniform $q_i$ and effective radius seems significant, since observations show that this is typically not the case. Uniformity is certainly the most simple approach, and prescribing some vertical structure would also involve big assumptions, so I agree with the authors' approach here. And while adding some vertical structure would affect the absolute values of CRH, my

intuition is that it would not have a big qualitative impact on the interscheme differences. However, I think a brief discussion of this is probably warranted, either in the Methods or Discussion section.

3. The explanations of interscheme differences focus on three parameters: the absorption coefficient, asymmetry parameter $g$, and SSA. It would be helpful to clarify the impact of ice crystal habit assumptions on these three parameters early in the paper, perhaps when they are first introduced around line 50 or when the schemes are compared in section 2.1. The sentence on line 169 implies that habit assumptions manifest through the $g$ – I am not sure if it is also supposed to imply that habit has little impact on $K\_abs$. Either way, it might be useful for many readers to have a couple of sentences early on that explicity connect the dots between habit assumptions and these three important parameters.

**Line comments**

Line 58 -   I thought this sentence was saying that the global ice CRE was somehow positive when surface roughness was taken into account. I checked the citation and realized the 2 W/m2 was *relative* to the case without surface roughness. This could be clarified with wording such as "...roughness, which can increase the global SW CRE by 2 W/m2..."

Line 108 – while it's nice to see a citation, Sokol et al (2024) did not feature any RTM simulations

Table 2 – the meaning of the Temperature column is specified for tests 2a and 2b, but not for the other tests. I think these are cloud top temperatures, but it would be good to specify.

Lines 135-136 – I am curious to what extent the lower air density can explain the increase in maximum heating rate as the cloud is moved upward. There may also be a contribution from the increasing difference in temperature between the lower-tropospheric emission temperature and the ice cloud temperature. Quantifying the impact of the density change would be relative straightforward using Eq 1, converted to height units so that density appears explicity.

Line 137 – I am confused here about q_i being the mixing ratio *per volume air*. I understand the point being made, but I find this wording confusing.

Line 146 – here co-albedo is defined as the "amount of absorbed to scattered radiation". Shouldn't this be the ratio of absorbed to total attenuated (absorbed+scattered) radiation?

Line 178/182: is it more accurate to say that the Baran schemes show stronger SW heating at the smallest cloud thicknesses (as opposed to the highest altitudes)? For most of Fig 3, the opposite is true, with weaker heating in the Baran schemes regardless of altitude. Is the

dependence of this result on cloud thickness just a reflection of the underlying dependence of SSA and $g$ on q_i?

Line 183: again, useful to specify that the "stronger absorption than in Fu" is only at the very low end of cloud thicknesses.

Line 212: **a**cross

Line 213: "10% less cloud-top cooling than the Fu scheme".  Fig 4h shows negative values at cloud top for the deep clouds. Doesn't this mean *more* cloud-top cooling than Fu?

Line 224: "show **o** in-cloud LW…"

Line 234: excepting -> except

Line 238: Does the trend really disappear below an IWP of 10 g/m2? Or is it just that cloud-top CRH in the Fu scheme changes sign? It seems the general trend of weaker CRH values (be they positive or negative) still applies below 10 g/m2.

Line 290: based on the earlier discussions, doesn't forward scattering act to increase CRH? As written here, it seems to suggest the opposite. Maybe the wording could be adjusted to make clear that it is the absorption capacity that wins out in spite of the increase in forward scattering?

---

## Author Response (AR1)

**Author's response**

(1) **In red: review comments**
(2) **In black: author's response**
(3) **In blue: author's changes in revised manuscript**

I would like to thank the referees for their feedback, reading effort and time dedicated on reviewing the manuscript proposed for submission to ACP journal.

**Referee 1 (R1), broad comment:**

"This paper by Sepulveda Araya et al compares ice-cloud radiative heating rates derived from four different radiative transfer schemes. The paper is very well motivated, well organized, and well written. The analysis is sufficiently thorough to explain why the different schemes produce different heating rates, and the presentation and discussion are very clear. The results of the intercomparison are quite nuanced, but there are two primary points: (1) accounting for ice crystal complexity generally weakens CRH, and (2) temperature-dependent schemes can lead to large CRH differences at cold temperatures characteristic of the upper troposphere.

The paper is well suited for ACP and will appeal not only to readers in the ice microphysics, cloud-radiation, and cloud-climate communities. I learned a lot from the paper, and think it has several valuable lessons for folks such as myself, who care about the role of ice clouds in climate but have little technical knowledge of the radiative transfer schemes on which our work relies.

The paper could be published as is, with some technical corrections. Nevertheless, I had a couple of general suggestions and number of line comments (mostly just clarification), so I will suggest minor revisions in case the authors wish to incorporate any of it. I look forward to seeing the paper published.

- Adam Sokol"

**Author's response:** Thanks for kind words. And thanks to bringing this, since one of the purposes of this work is to give a practical guidance and a sense of the impact of ice optical parameterizations in atmospheric numerical calculations.

**R1 general comment 1:**

"The figures and analysis all focus on vertically resolved CRH which, for reasons discussed in the Intro, is important for many reasons. Layer-averaged CRH may provide an additional, simpler perspective on interscheme differences. In Fig 2b,e, for example, showing layer-averaged CRH would indicate whether the positive and negative lobes of the dipole cancel out, or whether the net effect is nonzero. This could help qualify which downstream impacts may be affected by interscheme differences; for example, a layer-averaged difference near 0 wouldn't be so important for planetary-scale circulation, but ~10 K/day certainly would be.

For Fig S8 showing the layer-averaged CRH would convey whether the total heating is dependent on cloud thickness, or if the total is invariant but distributed differently throughout the cloud as the thickness varies. If the schemes produced different results in this regard, I would think that would be quite interesting.

Another option (closely related to the layer-averaged heating) would be to show the TOA CRE on top of each heating matrix. The CRH differences suggest that the TOA CRE differences would be very large in some of these cases."

**Author's response:** The layer-averaged CRH is an interesting and useful information to infer from the matrix visualization indeed. We think the colors in the heating rate matrices already give an idea if heating or cooling dominates through the column. Furthermore, to give a sense of the CRH distribution through the cloud, the maximum and minimum values are represented in the color bar as yellow lines, which already show whether the positive or negative CRH is greater. For example, in Fig2b, the absolute positive and negative values of the CRH are approximately the same (less than 1 K d$^{-1}$ difference). The color distribution in the matrix visualization shows the red/blue dipole in each column, resulting then in a layer-average CRH of approximately 0. In Fig2h, the positive CRH value is ~10 K d$^{-1}$ and the negative value is ~5.5 K d$^{-1}$. These absolute maximums result on a positive layer-averaged CRH.

However, to include the direct inference on the cloud-average CRH suggested, we will include three figures instead of including the numbers at the top row of each of the actual matrix visualizations in the preprint, one **after** actual Figure 2 of the preprint (for Test 1, cloud temperature), and the corresponding Test2a (cloud bottom temperature) and Test2b (cloud top temperature) figures in the supplement. These new figures show the CRH cloud-averaged CRH trends for each scheme vs the cloud parameter that is been tested.

**Author's changes: New paragraph included in line 185 of revised manuscript, describing the new figure 5:**

"To summarize the overall atmospheric impact of each optical scheme, Fig. 5 shows the cloud-average radiative heating rate ($\overline{CRH}$), computed as:

$$\overline{CRH} = \int CRH(z) q_i \rho_a dz$$

Optical schemes such as Baran14 and Yi13, both result in weaker $\overline{CRH}^{SW}$ relative to Fu, following the trend discussed above. However, no difference in Baran14 $\overline{CRH}^{SW}$ is found at cold clouds, showing the limitations of a not vertically-resolved CRH interscheme difference. Equally stronger in-cloud and weaker clod-top heating results in net zero vertically integrated $\Delta\overline{CRH}^{SW}$. Both Baran14 and Yi13 show a uniform weakening of 5% in $\overline{CRH}^{LW}$ through all temperature ranges. The inversion in Baran16 LW CRH mentioned in the previous paragraph can be found again, where $\overline{CRH}$ in all three components changes from negative values to a CRH enhancement relative to Fu, at colder levels. While some features of the CRH interscheme difference can be detected in an average number, which in turn can be useful for satellite comparison at the top of the atmosphere (TOA), the vertically resolved CRH difference is a key component when evaluating other atmospheric mechanisms, as radiative induced vertical motion. Radiative-related cloud processes cannot be completely understood with only a cloud-average radiative heating rate analysis."

[Figure]

**Figure 5. Ice optical schemes with higher complexity results in considerable weaker Cloud-average SW CRH.** SW (a), LW (b) and Net (c) Cloud-average CRH (y-axis) evaluated for each single-column profile shown in Fig. 2 across cloud temperatures from 201 to 236 K in Test 1 (x-axis). All four ice optical schemes under test are shown from darker blue to light blue, indicating low to high ice complexity level.

**Author's changes: New sentence included in line 244 of revised manuscript:**
"Similar to Fig. 5, the $\overline{CRH}$ for Test 2a and 2b are shown in Fig. S9 and S12 of the supplement, respectively."

**New Figures S9 and S12 in the supplement:**

[Figure]

**Figure S9.** SW (a), LW (b) and Net (c) Cloud-average CRH (y-axis) evaluated for each single column profile shown in Fig. S7. Cloud bottom temperatures across cloud depths from 0.5 to 5 km, with fixed cloud top temperature of 203 K (Test 2a) are depicted in x-axis. All four ice optical schemes under test are shown from darker blue to light blue, indicating low to high ice complexity level.

[Figure]

**Figure S12.** SW (a), LW (b) and Net (c) Cloud-average CRH (y-axis) evaluated for each single column profile shown in Fig. S9. Cloud top temperatures across cloud depths from 0.5 to 5 km, with fixed cloud bottom temperature of 237 K (Test 2b) are depicted in x-axis. All four ice optical schemes under test are shown from darker blue to light blue, indicating low to high ice complexity level.

**R1 general comment 2:**

"The assumption of vertically uniform q_i and effective radius seems significant, since observations show that this is typically not the case. Uniformity is certainly the most simple approach, and prescribing some vertical structure would also involve big assumptions, so I agree with the authors' approach here. And while adding some vertical structure would affect the absolute values of CRH, my intuition is that it would not have a big qualitative impact on the interscheme differences. However, I think a brief discussion of this is probably warranted, either in the Methods or Discussion section."

> **Author's response:** This is a key point. We decided to use the condensed matrix visualization to study how the CRH respond to specific parameter perturbations, when keeping other parameters constant. This method allows us to focus only on changes in $q_i$ or in $r_e$ independently and helps to develop an idea on what the impact would be in a more realistic scenario. For instance, and from a practical point of view, when finding huge difference in CRH when running atmospheric simulation, we can have a hint on which parameter is causing the biggest impact: the optical scheme assumed, qi profile, cloud height position or the effective radii retrieved from the microphysical scheme.
>
> We are currently doing a similar analysis by running storm-resolving models to continue studying the impacts of optical schemes and ice complexity on CRH. While those preliminary results will be discussed in an upcoming paper, the general trend of CRH weakening when including ice complexity can be found again, showing not a huge qualitative impact on the results, although, with some specific scenarios to be studied in detail. As an example, Figure 1a shows some of the **domain average** CRH vertical profiles retrieved when running ICON storm-resolving model+ecRad over the Asian monsoon region, where schemes such as Baran16 and Yi13 results on different levels of CRH weakening. However, the preliminary result showed in Figure 1a requires analysis on specific subdomain-timesteps to evaluate the contribution of specific cloud-scenarios, instead of focusing on the average-CRH. This is an example of potential workflow that can incorporate the matrix visualization shown in this paper: to evaluate CRH in more realistic scenarios, we can include analysis on the effective radius used by the optical schemes, and also, on specific variations on qi profile, to develop

an argument on which parameter is making an specific optical scheme to result in such CRH profile.

[Figure]

**Figure AC1.** Average net-CRH profiles over the Asian monsoon region (a) and corresponding average qi profile (b). ICON v2.6.4 + ecRad were used to evaluate the different optical schemes in (a) and (b). Panel (c) corresponds to the heating rate matrix visualization taken from Figure5i of the current preprint under review. The altitude-IWP range scenario from the storm-resolving simulation in (a) and (b) is shown inside the square-box, with Net CRH differences roughly ranging from less than -0.1 to -1 K d⁻¹.

From a similar comment given by the reviewer #2 (see R2 comment 5), we decided to include a new paragraph at the end of the Discussion and Conclusion section.

**Author's changes: New paragraph included in line 355 of revised manuscript:**
"Although the idealized single-column tests used in this work assume a vertical uniform $q_i$ and $r_e$ distribution through the cloud layers, which is not a typical realistic case, the perturbation method allows us to focus on independent impacts of $q_i$ and $r_e$ on CRH profiles in the tropics. From an atmospheric modeler point of view, or for microphysics studies, the sensitivity tests showed here offers a hint on which parameter evaluated in this work, is causing the biggest impact on computed CRH. Cases where a thin high peak of the $q_i$ profile located at cold temperatures, result in the largest differences of the retrieved CRH by the variety of ice complexity assumption. Therefore, these cases should be analyzed and processed with caution when are present in forecast and climate models. Moreover, while we expect SZA to result on a variety of SW CRH structure, as it increases the chances of radiation to be absorbed when SZA is high, other parameters assumed as fixed in this test, such as SW albedo and LW emissivity, would have a systematic impact on CRH profiles, and therefore, the interscheme difference trend".

**R1 general comment 3:** "The explanations of interscheme differences focus on three parameters: the absorption coefficient, asymmetry parameter g, and SSA. It would be helpful to clarify the impact of ice crystal habit assumptions on these three parameters early in the paper, perhaps when they are first introduced around line 50 or when the schemes are compared in section 2.1. The sentence on line 169 implies that habit assumptions manifest through the g – I am not sure if it is also supposed to imply that habit has little impact on $K_{abs}$. Either way, it might be useful for many readers to have a couple of sentences early on that explicitly connect the dots between habit assumptions and these three important parameters."

**Author's response:** Thanks to this and a similar comment from R2 (see R2-comment2), we decided to include the optical properties differences between Yi13 and Fu, as a function of effective radius in section 2.1 (See Figure 1 in author's change below). This figure is a summary of the third column panels from Figures S6 and S7 of the preprint supplement. This would be used as a guide on what are the impacts of different ice crystal assumptions on $K_{ext}$, ssa, $K_{abs}$ and g. These relative differences are used in the result section to explain the CRH differences, so we think moving some of the Supplementary material figures to the main text would help the reader to follow the explanations. These trends had already been introduced in previous works such as Yi et al 2013 and Yi 2022, but we agree that including those figures in the main text here, will avoid the reader to check other publications or even the supplementary material. The description of the new Figure (Figure 1), what the optical properties are, why these optical properties are used, and the impact of ice complexity on the optical properties will be included as a new paragraph after line 90 of the preprint. By doing this and following the suggestion from R2-comment2, we expect that the "readers would have a better understanding of the optical properties before reading the sensitivity studies for better comprehension". We will also move the next sentences from section 3.1 to the new paragraph in section 2.1:

"The mass absorption coefficient $K_{abs}$ is computed as the product between the mass extinction coefficient and the co-albedo, or amount of absorbed to scattered radiation $(1 - \omega_0)$."
"$K^{SW}_{abs}$ is also 18% smaller in Yi13 than Fu for our effective radius in this test (30 μm) (Figs. S6i and S6g). This property difference results in weaker absorption in the Yi13 scheme."

**Author's changes: New paragraph included in line 86 of revised manuscript:**
"These optical schemes parameterize the mass extinction coefficient $K_{ext}$, single scattering albedo $\omega_0$, and asymmetry parameter *g* as a function of the inputs mentioned above. The mass extinction coefficient $K_{ext}$ is the result of both, mass absorption coefficient $K_{abs}$ and mass scattering coefficient $K_{sca}$. $K_{abs}$ is computed as the product between $K_{ext}$ and the co-albedo, or amount of absorbed to total attenuated radiation $(1 - \omega_0)$, then quantifying how much radiation is absorbed and how much is scattered through an ice cloud. Finally, *g* quantifies the amount of forward versus backward scattered radiation by ice crystals."

**New sentences included in line 97 of revised manuscript:**
"The interscheme difference in the optical properties between Yi13 and Fu is shown in Fig. 1. $K^{SW}_{abs}$ is 18% smaller in Yi13 than Fu for our effective radius in this test (30 μm), as a result of smaller $K^{SW}_{ext}$ and larger $\omega_0^{SW}$. In other words, the differences in optical properties result in weaker absorption in the Yi13 scheme. The dependence of optical properties on the effective radius for each scheme is shown in Figs. S1 and S2 of the supplement, for both SW and LW, respectively."

[Figure]

**Figure 1. A more complex ice optical scheme leads to weaker SW and LW absorption.** Broadband relative difference of mass extinction coefficient $K_{ext}$ (solid line), single scattering albedo $\omega_0$ (dashed line), mass absorption coefficient $K_{abs}$ (dotted line) and asymmetry parameter *g* (dotted-dashed line), between Yi13 and Fu optical parameterization, as a function of ice crystal effective radii $r_e$. Both radiative components, SW and LW, are shown in blue and red, respectively. Spectrally resolved and absolute values for each scheme are shown in Figs. S1 and S2 of the supplement.

**Author's response:** Regarding the next comment: "The sentence on line 169 implies that habit assumptions manifest through the g – I am not sure if it is also supposed to imply that habit has little impact on $K_{abs}$.". The reviewer is correct about the not completely accurate sentence of habit having little impact on $K_{abs}$. Baran16 scheme compile his ice crystal habit parameterization as a function of $q_i$ and temperature. To avoid confusion on this, we decided to change that sentence to: "Differences in $g^{LW}$ are much smaller in this case, indicating that a temperature-dependent $K^{LW}_{abs}$ has a stronger effect on the optical properties than a non-temperature dependent ice crystal habit description".

**Author's changes: New paragraph included in line 184 of revised manuscript:**
"Differences in $g^{LW}$ are much smaller in this case, indicating that a temperature-dependent $K^{LW}_{abs}$ has a stronger effect on the optical properties than a non-temperature dependent ice crystal habit description."

**Author's response:** Additionally, and as is mentioned in R2-comment2 response below, FigureS5 will be moved from the preprint supplementary material to the main text in section 3.1. This figure shows the interscheme optical properties difference between Baran14, Baran16 relative to Fu. As these ice optical schemes depends on non-fixed parameters, such as $q_i$ and temperature in Test1, we use the matrix visualization again to put the differences under context. This case corresponds to the explanation for Test1 differences and including Figure S5 here would guide the reader to understand the g and $K_{abs}$ discussion commented before. Once the reader has a sense on how the optical properties impact the CRH variety of magnitudes in Test1, will be possible to continue with the rest of tests analysis, now only checking the supplementary material when necessary.

**Author's changes: Figure S5 moved to main text, as Fig. 4:**

[Figure]

**Figure 4. The variety of** $K_{abs}$ **and** $g$ **gradient found for** $q_i$**- and** $T$**-dependent schemes helps to explain the CHR differences.** Matrix visualizations including 22 interscheme relative difference profiles for the $K_{ext}$, $\omega_0$, $K_{abs}$ and $g$ (top to bottom rows), each for middle cloud temperatures from 201 K to 236 K (Test 1). First and second column shows Baran14-Fu interscheme difference for SW and LW components respectively. The same components are shown in the third and fourth column for Baran16-Fu interscheme comparison.

**R1 line comments:**

**R1 comment:** Line 58 - I thought this sentence was saying that the global ice CRE was somehow positive when surface roughness was taken into account. I checked the citation and realized the 2 W/m2 was relative to the case without surface roughness. This could be clarified with wording such as "
...roughness, which can increase the global SW CRE by 2 W/m2... "

**Author's response:** Thanks for this important observation. We agree on clarifying the 2 W/m2 as a difference in SW CRE, and not as an absolute CRE. We will change then line 58 to: "...which can increase the global SW CRE by 2 W m−2 at TOA..."

**Author's changes: New paragraph included in line 58 of revised manuscript:**
"...which can increase the global SW CRE by 2 W m−2 at TOA..."

**R1 comment:** Line 108 – while it's nice to see a citation, Sokol et al (2024) did not feature any RTM Simulations

**Author's response:** Thanks for the clarification. Probably I was thinking on including "Sokol, A. B., & Hartmann, D. L. (2020). Tropical anvil clouds: Radiative driving toward a preferred state. Journal of Geophysical Research: Atmospheres, 125, e2020JD033107. https://doi.org/10.1029/2020JD033107" when writing that sentence and I got confused, as in that work there is a comment about including $r_e$ and IWC profiles into RRTMG. However, I finally used the numbers mainly from Hartmann and Berry, 2017, so we will remove Sokol et al (2024) from line 108, but we will include Sokol and Hartmann (2020) in line 110, as reference for the bottom limit on ice cloud geometrical thickness.

**Author's changes: Reference removed in line 121 of revised manuscript, and sentence included in line 124:**
"The geometrical depth range follows similar values found by Sokol and Hartmann (2020)"

**R1 comment:** Table 2 – the meaning of the Temperature column is specified for tests 2a and 2b, but not for the other tests. I think these are cloud top temperatures, but it would be good to specify.

**Author's response:** We will modify the second sentence in Table 2 caption to "The temperature values shown in the second column correspond to middle cloud temperature, except in Test 2a and Test 2b, where the numbers indicate cloud bottom and cloud top ranges, respectively."

**Author's changes: Last sentence of caption of Table 2 modified to:**
"The temperature values shown in the second column correspond to middle cloud temperature, except in Test 2a and Test 2b, where the numbers indicate cloud bottom and cloud top ranges, respectively."

**R1 comment:** Lines 135-136 – I am curious to what extent the lower air density can explain the increase in maximum heating rate as the cloud is moved upward. There may also be a contribution from the increasing difference in temperature between the lower-tropospheric emission

temperature and the ice cloud temperature. Quantifying the impact of the density change would be relative straightforward using Eq 1, converted to height units so that density appears explicitly.

**Author's response:** You are correct. While qi is the dominant cause in SW, stronger temperature difference between next-to-surface emission and cloud altitude is the dominant cause in LW CRH enhancement. We will change the sentence "Again, lower air density results in maximum heating rates of~35 K day$^{-1}$ for this simulation" to "While lower air density still explains the increase in CRH, strong temperature difference between a warm lower-tropospheric emission and cold ice cloud temperature results in maximum LW heating rates of ~35 K day$^{-1}$ for this simulation"

**Author's changes: Line 149 of revised manuscript**
"While lower air density still explains the increase in CRH, strong temperature difference between a warm lower-tropospheric emission and cold ice cloud temperature results in maximum LW heating rates of ~35 K day$^{-1}$ for this simulation"

**R1 comment:** Line 137 – I am confused here about qi being the mixing ratio per volume air. I understand the point being made, but I find this wording confusing.

**Author's response:** We will remove the sentence "The lower air density also causes qi to increase at the highest altitudes, for a fixed mass of ice, as qi is a mixing ratio per volume air" to avoid confusion and a redundant argument.

**Author's changes: Sentence** "The lower air density also causes qi to increase at the highest altitudes, for a fixed mass of ice, as qi is a mixing ratio per volume air" **in line 150 of revised manuscript is removed**

**R1 comment:** Line 146 – here co-albedo is defined as the "amount of absorbed to scattered radiation". Shouldn't this be the ratio of absorbed to total attenuated (absorbed+scattered) radiation?

**Author's response:** Good observation, you are correct. "scattered radiation" will be changed to "total attenuated radiation"

**Author's changes: Line 89 of revised manuscript corrected:**
"total attenuated radiation"

**R1 comment:** Line 178/182: is it more accurate to say that the Baran schemes show stronger SW heating at the smallest cloud thicknesses (as opposed to the highest altitudes)? For most of Fig 3, the opposite is true, with weaker heating in the Baran schemes regardless of altitude. Is the dependence of this result on cloud thickness just a reflection of the underlying dependence of SSA and g on qi?

**Author's response:** Yes indeed. We will modify the beginning of the sentence in line 178 of the preprint to "The exception is in thin clouds at high altitudes". We still include the high-altitude aspect to contrast with Test 2b, where we have thin clouds at low altitudes.
Yi13 scheme is a reference point to establish CRH difference based only on ice complexity and inclusion of more habits. Then, in Baran14 and Baran16 there's not only difference in SSA and

g based on ice crystal habits, but also on dependence on qi (and on T for Baran16 in particular), so we expect that the positive anomaly in SW CRH at thin clouds, which is a characteristic present only in Baran simulations, is a reflection on the input parameters dependence.

**Author's changes: Line 205 of revised manuscript corrected:**
"The exception is in thin clouds at high altitudes"

**R1 comment:** Line 183: again, useful to specify that the "stronger absorption than in Fu" is only at the very low end of cloud thicknesses.

**Author's response:** We agree with this suggestion. We will include "at cloud top layers" at the end of that sentence.

**Author's changes: Line 210 of revised manuscript corrected:**
"at cloud top layers"

**R1 comment:** Line 212: across

**Author's response:** We will correct that typo, removing the ".".

**Author's changes: Line 239 of revised manuscript corrected**

**R1 comment:** Line 213: "10% less cloud-top cooling than the Fu scheme". Fig 4h shows negative values at cloud top for the deep clouds. Doesn't this mean more cloud-top cooling than Fu?

**Author's response:** It is 10% more cloud-top cooling indeed. Baran16 tends to enhance the LW CRH dipole, by increasing both in-cloud heating and cloud-top cooling. Thanks for the observation. This will be corrected.

**Author's changes: Line 240 of revised manuscript corrected:**
"50% more in-cloud heating and 10% less cloud-top cooling than the Fu scheme"

**R1 comment:** Line 224: "show o in-cloud LW…"

**Author's response:** We will change "o" for "an".

**Author's changes: Line 251 of revised manuscript corrected:**
"…all four schemes show an in-cloud LW heating…"

**R1 comment:** Line 234: excepting -> except

**Author's response:** Thanks, we will apply the requested correction.

**Author's changes: Line 261 of revised manuscript corrected:**
"except at IWP greater than 100 g m$^{-2}$"

**R1 comment:** Line 238: Does the trend really disappear below an IWP of 10 g/m2? Or is it just that cloud-top CRH in the Fu scheme changes sign? It seems the general trend of weaker CRH values (be they positive or negative) still applies below 10 g/m2.

> **Author's response:** Good observation and good interpretation of the heating matrix visualization. In this case, there is a confusion on CRH weakening trend, and CRH vertical profile trend. We will modify the sentence in line 238 to "Importantly, this trend is maintained below an IWP of 10 g m$^{-2}$, although, with a different CRH vertical structure. For ice clouds with the lowest amounts of condensate, the cloud-top cooling becomes less important, and LW CRH differences are negative throughout the cloud depth."

> **Author's changes: Line 265 of revised manuscript modified:**
> "Importantly, this trend is maintained below an IWP of 10 g m$^{-2}$, although, with a different CRH vertical structure. For ice clouds with the lowest amounts of condensate, the cloud-top cooling becomes less important, and LW CRH differences are negative throughout the cloud depth."

**R1 comment:** Line 290: based on the earlier discussions, doesn't forward scattering act to increase CRH? As written here, it seems to suggest the opposite. Maybe the wording could be adjusted to make clear that it is the absorption capacity that wins out in spite of the increase in forward scattering?

> **Author's response:** The reviewer is correct. The relative weaker absorption capacity in complex ice schemes is the leading and primary cause to weaker CRH, and on which this sentence should be focused on. For the sza evaluated in this work, more forward scattering leads to chances that the SW radiation would be absorbed within the cloud, by reaching lower cloud levels. We expect that the lower the SZA, the greater the chances of SW radiation being transmitted, and therefore less absorbed by a thin cloud. However, as the sza is not a parameter test evaluated in this test, the potential impact of smaller differences in g and its relations with sza will be included in another paragraph (See response to R2-comment3)
> Following your suggestion then, we will modify this sentence to: "In general, when more ice crystal habits are included, the absorption capacity of ice clouds decreases greatly, while the forward scattering increases to a lesser extent, resulting in all weaker SW CRH, LW in-cloud heating, and cloud-top cooling."

> **Author's changes: Line 318 of revised manuscript modified:**
> "In general, when more ice crystal habits are included, the absorption capacity of ice clouds decreases greatly, while the forward scattering increases to a lesser extent, resulting in all weaker SW CRH, LW in-cloud heating, and cloud-top cooling."

> ========================================================================

**Referee 2 (R2), broad comment:** "The results are helpful to the readers in knowing the impacts of different ice property scheme on ice cloud CRH simulations. The topic is within the scope of ACP, and the paper is logically organized and clearly written. I mostly support the publication of this paper, but there are some suggestions for the authors and some concerns to be addressed."

**Author's response:** Thanks for the supportive words and highly constructive suggestions. I am glad the reviewer consider these results helpful to the reader and I appreciate the support for the publication of this paper.

**R2 comment 1:** "The authors may need to consider changing the title. The present title seems to be broad and comprehensive, but in fact, the present study has several limitations, such as idealized, single-column calculation, a few (not all) cloud properties, and for several optical schemes."

**Author's response:** Following the previous good observation point, we decided to change the article title to "Ice crystal complexity leads to weaker ice cloud radiative heating in idealized single-column simulations". Although limited to a sensitive experiment, most of all simulation results showed here are explained in terms of changes in $K_{abs}$ and in g. Knowing how $K_{abs}$ and g differs between optical scheme, based on ice crystal habits assumption, helps to have an idea on how CRH vertical profile can respond not only in this study, but also on different radiative transfer calculations, and in atmospheric modelling.

Moreover, this comment by R2, motivates us to follow a missing ACP guideline point about article titles, such as highlighting the main scientific results in the title (see https://www.atmospheric-chemistry-and-physics.net/policies/guidelines_for_authors.html).

**Author's changes: Title of manuscript:**
"Ice crystal complexity leads to weaker ice cloud radiative heating in idealized single-column simulations"

**R2 comment 2:** "It is suggested that the authors show more details (figures and discussions) about the four ice optical schemes used in this study, such as the mass extinction coefficients, single-scattering albedo, asymmetry factor, etc., in the main text.. so that the readers could have a better understanding of the optical properties before reading the sensitivity studies for better comprehension."

**Author's response:** Following this suggestion, and similar to R1-general-comment3 response, we decided to move some figures from the Supplementary material to the main text. As commented previously, the right column of Figures S6 and S7 will be included in section 2.1, as a new condensed figure (See Figure 1 in R1-general-comment3), which can help the reader to have a better understanding of the interscheme optical differences before getting into the results. This will help as an introduction on how ice crystal complexity impacts the optical properties of ice clouds. However, Figure 1 focused only on differences between Yi13 and Fu scheme, as both schemes depends only on effective radius $r_e$, making it easier for a quick comparison figure. The interscheme differences on optical properties for Baran 14 and Baran16 relative to Fu (Figure S5 of the prepint), will be moved from the supplementary material to section 3.1 of the main text. This figure continues giving an idea of the optical property differences, between different type schemes, but under the context of simulated clouds in Test 1. This can help the reader to follow the text and the explanation of CRH differences based on $K_{abs}$ and g.

**Author's changes: See new paragraphs included in author's changes from R1-general-comment3. This corresponds to changes in line 86 and 97 of revised manuscript.**

**New paragraph included in line 86 of revised manuscript:**
"These optical schemes parameterize the mass extinction coefficient $K_{ext}$, single scattering albedo $\omega_0$, and asymmetry parameter $g$ as a function of the inputs mentioned above. The mass extinction coefficient $K_{ext}$ is the result of both, mass absorption coefficient $K_{abs}$ and mass scattering coefficient $K_{sca}$. $K_{abs}$ is computed as the product between $K_{ext}$ and the co-albedo, or amount of absorbed to total attenuated radiation $(1 - \omega_0)$, then quantifying how much radiation is absorbed and how much is scattered through an ice cloud. Finally, $g$ quantifies the amount of forward versus backward scattered radiation by ice crystals."

**New sentences included in line 97 of revised manuscript:**
"The interscheme difference in the optical properties between Yi13 and Fu is shown in Fig. 1. $K^{SW}_{abs}$ is 18% smaller in Yi13 than Fu for our effective radius in this test (30 µm), as a result of smaller $K^{SW}_{ext}$ and larger $\omega_0^{SW}$. In other words, the differences in optical properties result in weaker absorption in the Yi13 scheme. The dependence of optical properties on the effective radius for each scheme is shown in Figs. S1 and S2 of the supplement, for both SW and LW, respectively."

**R2 comment 3:** "Although the authors have already done a lot of work to evaluate the sensitivities of ice cloud radiative heating to optical, macro- and microphysical properties, the results still seem to be limited given that many more key parameters are fixed here (line 100-105), such as the skin temperature, tropical oceans, SW surface albedo, LW surface emissivity, effective solar zenith angle (SZA), etc. How would the different settings of these parameters could possibly affect the validity and magnitude of the present results? I think the results would be even quite valuable if the authors could elaborate on this."

**Author's response:** We expect the sza to be an important parameter, as this could give a variety of results (See response to R1-L290comment), as the interscheme difference in asymmetry parameter g alters the scattering/absorption chances on cloud top level, if sza is high, but also determines how much radiation is transmitted through the cloud when sza is low, then affecting the CRH profile. Although out of the scope of this work already, we are currently working on diurnal cycles of the CRH using a storm-resolving model and the different optical schemes tested here. We expect this work to bring some hint on the diurnal cycle, or equivalently, the impact of sza in CRH interscheme differences.

However, although the rest of parameters and conditions chosen here follow tropical conditions, and in particular over sea surface, we would expect a systematic impact from surface characteristics. Higher surface LW emissivity, or higher surface skin temperature would lead to stronger LW CRH dipole due to higher gradient between surface upward flux and cloud-bottom and -top fluxes. Higher SW albedo from a land surface for instance would enhance SW CRH due to increase chances of upward SW radiation to be absorbed by cloud bottom layer. We would expect then that the intershceme differences in CRH would follow same trend and structure, but different magnitudes.

We will include these points, and others from R1general-comment2 and R2-comment5 into a new paragraph in the discussion section.

**Author's changes: New paragraph included in line 355 of revised manuscript (SAME PARAGRAPH MENTIONED IN R1general-comment2):**

"Although the idealized single-column tests used in this work assume a vertical uniform $q_i$ and $r_e$ distribution through the cloud layers, which is not a typical realistic case, the perturbation method allows us to focus on independent impacts of $q_i$ and $r_e$ on CRH profiles in the tropics. From an atmospheric modeler point of view, or for microphysics studies, the sensitivity tests showed here offers a hint on which parameter evaluated in this work, is causing the biggest impact on computed CRH. Cases where a thin high peak of the $q_i$ profile located at cold temperatures, result in the largest differences of the retrieved CRH by the variety of ice complexity assumption. Therefore, these cases should be analyzed and processed with caution when are present in forecast and climate models. Moreover, while we expect SZA to result on a variety of SW CRH structure, as it increases the chances of radiation to be absorbed when SZA is high, other parameters assumed as fixed in this test, such as SW albedo and LW emissivity, would have a systematic impact on CRH profiles, and therefore, the interscheme difference trend".

**R2 comment 4:** "The figure captions in this paper looks weird. The authors may need to explain what the figures are before discussing the details. For example, Figure captions of 1, 2, and 3."

**Author's response:** We included the main and most important takeaway from each figure first, to help the reader follow the most important results of the sensitivity tests when reading the captions. Next to the main takeaway in bold, the reader can get into the characteristics and traditional figure description. We decided to keep the figure captions as they are.

**R2 comment 5:** "The last but important thing that the readers would care about, is how well the simulations perform as compared to the reality? This again would be very helpful and valuable if the authors could elaborate on this question."

**Author's response:** This is an important and interesting point to include in the proposed manuscript. This is a sensitivity test with the purpose to give the reader -being an atmospheric modeler or stalillite retrieval user- an idea on the impacts of ice optics assumptions under different cloud scenarios. Although the idealized single colum simulations shown in this preprint follows typical tropical conditions, and especially over sea surface, there are a serie of assumptions included, such as vertical profile of $q_i$ and $r_e$, that help to build the sensitivity and perturbation approach followed. The perturbation range used for this parameters looks to offer a sensitivity range of CRH that can be useful for evaluation of more realistic scenarios. We would expect that atmospheric models such as storm-resolving simulations to have at least similar qualitative results, in terms of weakening of CRH when ice crystal complexity is included. For example, and to address more on the utilitiy of the sensitivirty test in a more realistic simiulation, we are currently working on storm-resolving simulations, together with CloudSat/CALIPSO data to evaluate the inclusion of ice complexity (See Figure AC1 and response to R1 general commen 2).

Future simulations and evaluations under more realistic scenarios are of course necessary, and these are our following steps indeed. This is a challenging but necessary task as there is no direct measurement of CRH or in-cloud fluxes measurements. Even satellite products

assume ice optical schemes and ice crystal descriptions when delivering CRH, $q_i$ or $r_e$ products, so caution have to be considered when comparing with "realistic" CRH profiles.

To complement on this important point, and the limitations of the current sensitivity test, we will include the following lines at the beginning of line 292 paragraph.

**Author's changes: New sentences included in line 321 of revised manuscript:**

"The sensitivity test presented here looks to show the impacts of ice optics assumptions under different cloud scenarios based on realistic ranges. However, existing evaluations result on similar trends to those derived in this test. Previous works shows a ..."

The rest of this paragraph brings preceding evaluations, some of those based on more realistic atmospheric conditions. Ren et al 2021 study result will be included also in line 297, as this was not included in the original preprint. Ren et al 2021 did a similar evaluation by using cloud characteristics from the western equatorial Pacific region, retrieved from satellite measurements and GEOS-5 reanalysis data set. While they used RRTM-G radiative transfer scheme, and other ice complex optical parameterization relative to Fu with other optical characteristics, they found similar weakeking trend for the LW-CRH when including aggregates and surface roughness, which are similar habit characteristics to Baran14 and Baran16 scheme. As commented in line 298, the common trend is found in relative big scale systems simulations (Keshtgar et al 2024) and in more realistic climate models (Zhao et al 2018).

Also, we will include the next two paragraphs, focused on the limitations, purpose of this sensitivity study, and future work (same paragraph included in author's response to R1general-comment 2, and R2-comment3).

**Author's changes: New paragraph included in line 355 of revised manuscript (SAME PARAGRAPH MENTIONED IN R1general-comment2 and R2 comment3):**

"Although the idealized single-column tests used in this work assume a vertical uniform $q_i$ and $r_e$ distribution through the cloud layers, which is not a typical realistic case, the perturbation method allows us to focus on independent impacts of $q_i$ and $r_e$ on CRH profiles in the tropics. From an atmospheric modeler point of view, or for microphysics studies, the sensitivity tests showed here offers a hint on which parameter evaluated in this work, is causing the biggest impact on computed CRH. Cases where a thin high peak of the $q_i$ profile located at cold temperatures, result in the largest differences of the retrieved CRH by the variety of ice complexity assumption. Therefore, these cases should be analyzed and processed with caution when are present in forecast and climate models. Moreover, while we expect SZA to result on a variety of SW CRH structure, as it increases the chances of radiation to be absorbed when SZA is high, other parameters assumed as fixed in this test, such as SW albedo and LW emissivity, would have a systematic impact on CRH profiles, and therefore, the interscheme difference trend."

**Author's changes: New paragraph included in line 364 of revised manuscript:**

"As discussed previously, preliminary work including more realistic contexts, have already shown similar qualitative trends on CRH when ice complexity is included. However, more evaluations, by using storm-resolving models for instance, under specific cloud structures and characteristics will help to assess the utility and performing of the sensitivity test presented here. Including remote sensing products in future analysis is also a necessary part of the evaluation workflow, but caution must be considered as even satellite products assume ice crystal characteristics when retrieving CRH, $q_i$ or $r_e$ profiles (Wu et al. 2024, Ren et al 2023, Delanoë and Hogan, 2010)"

**Other author comments:**

From Anthony Baran's email on October 23, 2024. The email's author agrees on bringing his comment into the discussion:

"...My specific concern is about the contents of Table 1. In that table, you state that the Baran14 and 16 studies excludes surface roughness. This is not correct, surface roughness is including in the short-wave, between 0.2 and 4.9 microns. Both Baran14 and 16 utilize the same basic single-scattering properties of the ensemble model as described in this paper Baran, A. J., R. Cotton, K. Furtado, S. Havemann, L.-C. Labonnote, F. Marenco, A. J. Smith, and J.-C. Thelen, 2014a: A self-consistent scattering model for cirrus. II: The high and low frequencies. Quart. J. Roy. Meteor. Soc., 140, 1039–1057, doi:10.1002/qj.2193, which you do cite. It is true to say that surface roughness is not included in the infrared, between 5 and 120 microns..."

**Author's response:** "The surface roughness point is mentioned in Table 1 and two sentences in the manuscript where Yi13 and your schemes are described. Fortunately, we focused our analysis and results on scheme differences such as "general" ice crystal complexity and input parameters. I was trying to analyze surface roughness impact on the discussion, but this is a characteristic difficult to isolate from the rest of differences between schemes, even more now that you bring this comment." (taken from response sent to Anthony Baran by email)

Following this important point from Anthony Baran, we will apply the next changes:

**Author's changes: New sentence included in line 71 of revised manuscript:**
"... as temperature dependence (Baran et al. 2016). Both Baran's schemes prescribe surface roughness influence in the SW component. These..."

**Author's changes: Correction of Table 1 of revised manuscript:**
"No" is changed from No to a check mark and "(only in SW)" in Baran14 and Baran16 rows.

**Author's changes: New sentence included in line 103 of revised manuscript:**
"... ice mass mixing ratio. Surface roughness is included in the SW parameterization. The bulk..."

**Author's changes: New sentence included in line 310 of revised manuscript:**

"The Baran schemes use a simpler ensemble of ice crystal habits, together with surface roughness and Yi13 a more elaborate one, including hollowness and surface roughness properties"

**General author's changes: All Figures references in the text, from both main and supplement, have been updated.**

**References:**

Ren, T., D. Li, J. Muller, and P. Yang, 2021: Sensitivity of Radiative Flux Simulations to Ice Cloud Parameterization over the Equatorial Western Pacific Ocean Region. J. Atmos. Sci., 78, 2549–2571, https://doi.org/10.1175/JAS-D-21-0017.1.

Wu, B., Wang Y, Fan X., Liu S. and Fu Y, 2024: A mixing scheme of ice particle models for global ice cloud measurements. Remote Sensing of Environment, 313, 114356, https://doi.org/10.1016/j.rse.2024.114356

Ren, T., Yang, P., Loeb, N. G., Smith, W. L., Jr., & Minnis, P. (2023). On the consistency of ice cloud optical models for spaceborne remote sensing applications and broadband radiative transfer simulations. Journal of Geophysical Research: Atmospheres, 128, e2023JD038747. https://doi.org/10.1029/2023JD038747

Delanoë, J., and R. J. Hogan (2010), Combined CloudSat-CALIPSO-MODIS retrievals of the properties of ice clouds, J. Geophys. Res., 115, D00H29, https://doi:10.1029/2009JD012346

---

## Author Response (AR2)

**Author's response to editor:**

(1) **In red: editor's comments**
(2) **In black: author's response**
(3) **In blue: author's changes in revised manuscript**. Strikethrough text is used to show removed text from the manuscript and underline text is used to show new text added to the manuscript.

We would like to thank the editor for his revision and the corresponding time dedicated to it. We also appreciate his positive decision for publication. We address his comments below in this document.

"…I agree with Referee #2's suggestion to remove the "takeaway messages" from figure captions…"

> **Author's response:** Following the suggestion from Referee #2, the editor and ACP guidelines for figure captions, we decided to remove the "takeaway messages" from figure captions.
>
> **Author's changes: Bold "takeaway messages" removed from captions of Figures 1 to 9:**
>
> **P5 of marked-up manuscript:** 'Figure 1.  Broadband…'
> **P7 of marked-up manuscript:** 'Figure 2.  22 CRH profiles…'
> **P8 of marked-up manuscript:** 'Figure 3.  Heating rate…'
> **P9 of marked-up manuscript:** 'Figure 4.  Matrix visualizations…'
> **P10 of marked-up manuscript:** 'Figure 5.  SW (a), …'
> **P12 of marked-up manuscript:** 'Figure 6.  Heating rate matrix…'
> **P13 of marked-up manuscript:** 'Figure 7.  Heating rate matrix…'
> **P15 of marked-up manuscript:** 'Figure 8.  Heatin rate matrix…'
> **P17 of marked-up manuscript:** 'Figure 9.  Heating rate matrix…'

"l 30: 'CRH rather than CRE' you might consider replacing 'rather than' by 'besides'"

> **Author's response:** We replaced 'rather than' by 'besides'.
>
> **Author's changes:**
> **P2-L30 of marked-up manuscript:** 'There are a number of motivations to study CRH  besides CRE.'

"Table 1: Consider removing the star for Baran14* (there is no direct reference to that star in the caption)"

**Author's response:** We removed the star from Baran14* in Table 1.

**Author's changes:**
**P6- Table 1 of marked-up manuscript:** 'Baran14'.

"You may also consider rephrasing the entries of the 3rd column in: - no roughness, Both SW and LW and SW only"

**Author's response:** We rephrased the entries of the 3rd column in Table 1, following editor suggestions.

**Author's changes: P6- Column 3 in Table 1 of marked-up manuscript:**

| *Surface Roughness* |
| --- |
| ✘ No roughness |
| ✔  SW only |
| ✔ Both SW and LW |
| ✔  SW only |

"Table 2, caption: 'middle cloud temperature' -> 'mid-cloud temperature' ?"

**Author's response:** Thanks for noticing this. We replaced 'middle cloud temperature' to 'mid-cloud temperature' not only in Table 2 caption but also in Figure 4 caption.

**Author's changes:**
**P6-Table 2 caption of marked-up manuscript:** 'The temperature values shown in the second column correspond to  mid-cloud temperature...'
**P9-Figure 4 caption of marked-up manuscript:** '...each for  mid-cloud temperatures from 201 K to 236 K (Test 1)...'

"Lin 185: 'Fig. 5' should be spelt out 'Figure 5' as it is the subject of the sentence"

**Author's response:** Thanks for reminding us of this guideline. We changed 'Fig. 5' to 'Figure 5'.

**Author's changes:**
**P11-L181 of marked-up manuscript:** ' Figure 5 shows the cloud-average radiative heating rate...'

"The quantity computed using Eq. 3 is not an average, but a mass integral of the CRH, and is not consistent with the units of K day-1 shown in Fig. 5. Shouldn't it be normalized by the mass of air (or of ice)"

**Author's response:** Thanks for noticing this important mistake. We have updated Eq. 3 in the manuscript, using now Eq. AR1 (see below), to correctly evaluate the CRH averaged over cloud layers only. We had used this equation to calculate SW CRH (Figure 5a, Figure S9a, and Figure S12a). However, we mistakenly used the incorrect Eq. 3 to evaluate both LW and

net CRH (Figure 5b and c, Figure S9b and c, Figure S12 b and c). We have updated those LW and net CRH figure panels with the correct equation:

$$\overline{CRH} \ = \ \frac{\int_z CRH\ q_i\rho_a dz}{\int_z q_i\rho_a dz} \tag{AR1}$$

Other minor editions were applied to each figure 5, S9 and S12, as limits in the y-axis, position of panel labels and distribution of text legend. Calculations were performed in python. If the editor deems it necessary, the jupyter notebook corresponding to this calculation can be seen in the updated public repository https://github.com/EdgardoSepulveda/ice-crh-1column (Cell 7 under "Calculation of cloud-averaged CRH:" in files "3-figures_test1_v7b.ipynb", "3-figures_test2a_v7b.ipynb" and "3-figures_test2b_v7a.ipynb")

Additionally, paragraph starting from L187 has been updated. While absolute magnitudes of $\overline{CRH}$ changed in the LW and net radiative component, the trends, differences between schemes, and main takeaway from the figure are the same.

**Author's changes:**
**Figure 5 change:**
**OLD Figure 5**

**NEW Figure 5**

[Figure]

**Figure S9 change:**
**OLD Figure S9**

[Figure]

[Figure]

[Figure]

**NEW Figure S9**

[Figure]

[Figure]

[Figure]

**Figure S12 change:**
**OLD Figure S12**

[Figure]

[Figure]

[Figure]

**NEW Figure S12**

[Figure]

[Figure]

[Figure]

**Equation 3:**

$$\overline{CRH} = \int_z CRH\, q_i \rho_a dz \qquad\qquad \overline{CRH} = \frac{\int_z CRH\, q_i \rho_a dz}{\int_z q_i \rho_a dz}$$

**P11-L183, 184 of marked-up manuscript:** $\overline{CRH^{SW}}$ SW $\overline{CRH}$

**P11-L184 of marked-up manuscript:** '...showing the limitations  when analyzing a not vertically-resolved CRH'

**P11- L184 of marked-up manuscript:** weaker  cloud-top heating

**P11- L185 of marked-up manuscript:** $\overline{\Delta CRH^{SW}}$ SW $\Delta\overline{CRH}$

**P11-L186 of marked-up manuscript:** $\overline{CRH^{LW}}$ LW $\overline{CRH}$

**P11-L187 of marked-up manuscript:** '...uniform weakening of 5% in LW $\overline{CRH}$  throughout all temperatures ranges'

**P11-L187 of marked-up manuscript:** ' However, it is not possible to detect the inversion in Baran16...'

**P11-L188 of marked-up manuscript:** '...in the previous paragraph  with Baran16 LW $\overline{CRH}$  always being higher than the other schemes. In contrast, Baran16 Net $\overline{CRH}$ changes from lower to higher values than the other schemes, mainly due to the contribution of the SW component. While some features of the CRH...

"l210: 'at cloud top layers' -> 'at cloud top' ?"

**Author's response:** Thanks for this suggestion. We corrected this sentence to avoid confusion, and also to follow referee#1 comment ("again, useful to specify that the 'stronger absorption than in Fu' is only at the very low end of cloud thicknesses")

**Author's changes:**
**P11-L209 of marked-up manuscript:** 'the temperature-dependent $K_{abs}^{SW}$ causes stronger absorption than in Fu,  at the top of thin clouds.'

**Additional changes:**

In addition to the editor's comments, we included the next minor changes:

**P1-L18 of marked-up manuscript:** '...determines  temperature , pressure gradients...'

**P2-L25 of marked-up manuscript:** '...satellite  and ground-based  measurements...'

**P3-L66 of marked-up manuscript:** 'calculations' included in '...radiative transfer calculations and a variety...'

**P3-L72 to L74, P10-L165 of marked-up manuscript:** 'Section' to 'Sect.', following https://www.atmospheric-chemistry-and-physics.net/submission.html#manuscriptcomposition guidelines.

**P3-L80 of marked-up manuscript:** 'RRTM', as RRTM is the general radiative transfer model on which RRTM-G is based.

**P4-L98 of marked-up manuscript:** 'the effective radius tested in most of our simulations' and ', as described in Sect 2.2' included in '...Fu for  the effective radius tested in most of our simulations (30 μm, as describes in Sect. 2.2)...'

**P4-L120 of marked-up manuscript:** 'tropical' included in '... corresponding tropical temperatures'

**P5-Figure 1 and Figure 1 caption, Table 1 and Table 1 caption, P19 L354, L356 and L368 of marked-up manuscript:** '$r_e$' replaced by '$r_{eff}$' to be consistent in all manuscript.

**P5-L127 of marked-up manuscript:** 'direct' included in '...have no direct dependence on ice crystal...'

**P5-L130 of marked-up manuscript:** '...by fixing the IWP  and calculating...'

**P5-L131 of marked-up manuscript:** 'A fix IWP of 30 g m$^{-2}$ is used in Tests 1, 2 and 4. The IWP depends on qi...'

**P6-Table2 caption of marked-up manuscript:** 'temperature' included in '...cloud bottom and cloud top temperature ranges...'

**P8-Figure 3 caption of marked-up manuscript:** 'from least to more  complex'.

**P8-Figure 3 caption of marked-up manuscript:** new sentence included: 'As in Fig. 2, yellow lines in the color bar indicate the corresponding ΔCRH range for each panel'.

**P10-L180 of marked-up manuscript:** '...has a stronger effect  than a non-temperature...'

**P17-L309 of marked-up manuscript**: '...simpler ensemble, of ice crystals habits including surface roughness in the SW calculations, and Yi13 a more elaborate one, including hollowness and surface roughness properties in both SW and LW parameterizations.'

**P18-L354 of marked-up manuscript:** ' The idealized single-column tests...'

**P18-L355 of marked-up manuscript**: 'However, the perturbation...'

**P18-L357 of marked-up manuscript**: '...the sensitivity tests  shown here offer...'

**P18-L361 of marked-up manuscript**: '...as it  changes the chances of radiation...'

**P19-L363 of marked-up manuscript**: '...and therefore on the interscheme...'

**P19-L369 of marked-up manuscript:** Updated github link from private repository to public repository:  https://github.com/EdgardoSepulveda/ice-crh-1column.